# Quantifying the Impacts of Climate Change and Vegetation Variation on Actual Evapotranspiration Based on the Budyko Hypothesis in North and South Panjiang Basin, China

**Tiansheng Li [1], Jun Xia [1,2,3,\*], Dunxian She [1], Lei Cheng [1], Lei Zou [3] and Bojun Liu [4]**

[1]   State key laboratory of Water Resources and Hydropower Engineering Science, Wuhan University, Wuhan 430072, China; li_tiansheng@whu.edu.cn (T.L.); shedunxian@sina.com (D.S.); Lei.Cheng@whu.edu.cn (L.C.)

[2]   Hubei Key Laboratory of Water System Science for Sponge City Construction, Wuhan University, Wuhan 430072, China

[3]   Key Laboratory of Water Cycle and Related Land Surface Processes, Institute of Geographic Sciences and Natural Resources Research, Chinese Academy of Sciences, Beijing 100101, China; leiz@igsnrr.ac.cn

[4]   Yellow River Engineering Consulting Co., Ltd., Zhengzhou 450003, China; bojun_l689@126.com

\*   Correspondence: xiajun666@whu.edu.cn; Tel.: +86-027-6877-2215

**Abstract:** Actual evapotranspiration ($E_a$) plays a key role in the global water and energy cycles. The accurate quantification of the impacts of different factors on $E_a$ change can help us better understand the driving mechanisms of $E_a$ in a changing environment. Climate change and vegetation variations are well known as two main factors that have significant impacts on $E_a$ change. Our study used three differential Budyko-type equations to quantify the contributions of climate change and vegetation variations to $E_a$ change. First, in order to establish the relationship between the parameter n, which usually presents the land surface characteristics in the Budyko-type equations, with basic climatic variables and the Normalized Difference Vegetation Index (NDVI), the stepwise linear regression method has been used. Then, elasticity and contribution analyses were performed to quantify the contributions of different numbers of climatic factors and NDVI to $E_a$ change. The North and South Panjiang basin in China was selected to investigate the efficiency of the modified Budyko-type equations and quantify the impacts of climate change and vegetation variations on $E_a$ change. The empirical formal of the parameter n established in this study can be used to simulate the parameter n and $E_a$ for the study area. The results of the elasticity and contribution analyses suggest that climate change contributed (whose average contribution is 149.6%) more to $E_a$ change than vegetation variation (whose average contribution is −49.4%). Precipitation, NDVI and the maximum temperature are the major drivers of $E_a$ change, while the minimum temperature and wind speed contribute the least to $E_a$ change.

**Keywords:** actual evapotranspiration; Budyko hypothesis; climate change; vegetation variation; North and South Panjiang basin

---

## 1. Introduction

Driven mainly by solar radiation, terrestrial actual evapotranspiration ($E_a$) plays a key role in the water and energy exchange between land surface and atmosphere. In the context of global change, the accurate quantification of $E_a$ and identification of different factors (such as climatic factors and vegetation conditions) that influence $E_a$ change are crucial for evaluating the impacts of changes in

climate and vegetation on water and energy cycles, improving the accuracy of hydrological forecasting and developing a suitable strategy for water resource management.

Recently, many studies have indicated significant change in $E_a$ due to climate change based on statistical analyses [1–5]. These findings have increased the interest in investigating the potential climatic driving factors that control the changes in $E_a$. Some previous studies indicated that the impacts of changes in different climatic factors on $E_a$ change are quite different. For example, a change in temperature can affect $E_a$ primarily by changing the capacity of air to hold water vapor. Increased cloudiness and decreased solar irradiance can induce reductions in $E_a$ [6]. The decrease in solar irradiance was found to be the main cause of the decline in $E_a$ across many regions of the world [7,8]. As an important factor related to aerodynamic turbulence in $E_a$ calculations, wind speed affects $E_a$ by altering the process of vapor removal, which transfers large quantities of air over the evapotranspiration surface [6]. In addition to these statistical analyses, many mathematical models of $E_a$ estimation, such as the Penman–Monteith model [9,10], the Budyko hypothesis model [11–13] and the complementary relationship framework [14–16], have been used to analyze the effects of changes in various climatic factors on $E_a$ change.

However, an increasing number of studies have found that climate change is not the only driver of $E_a$ change [5,17,18]. Among the potential influential factors other than climate change, land surface vegetation conditions, which are the largest contributor to total evapotranspiration [19,20] and partly control the maximum soil water available for evapotranspiration, have received considerable attention in analyses of $E_a$ change in recent years [12,21–23]. In studies of the impacts of vegetation variations on $E_a$ change, Shao et al. [24] and Wang and Dickinson [9] found that afforestation contributes to $E_a$ increase at the mean annual scale and plants' growth increases $E_a$ at the seasonal scale. Jaramillo et al. [25] indicated that a main increase of $E_a$ is mainly due to the increase of cultivated area and/or crop production in Sweden. Zhang et al. [17] found that the significant upward trend of global $E_a$ from 1982 to 2013 was mainly driven by vegetation greening. Previous studies have suggested that it is necessary to explicitly incorporate key indices of vegetation conditions into the Budyko framework for many catchments [22,25–28]. Consequently, it is meaningful and important to investigate how these climate and vegetation characteristics collectively influence the spatial and temporal change in $E_a$.

Generally, the Budyko framework [11] has been widely used to investigate the impacts of climate change and vegetation variations on $E_a$ change in various catchments with different climate and land surface conditions because of its conceptual appeal and the fact that it requires only routinely recorded weather data [25,27–33]. In addition to the energy and water supplies in the Budyko framework, other factors can influence the process by which water escapes from the land surface into the atmosphere [11,22,28,34], and these factors can be numerically represented by the free parameter, n, in Budyko-type equations [35–37]. Moreover, many studies have found that the value of n is basin-specific [12,22,30] and suggested that it is necessary to better characterize the variables that contribute to the change in n. Based on previous studies, land surface characteristics, including climatic factors, vegetation conditions, topography and soil properties, have been identified as the potential descriptors of a catchment that control the change in n [12,21,22,28,30,33,38]. The influence of these physical characteristics on n is known to be important, and many researchers have explored how these physical characteristics collectively influence the spatial and temporal change in n [12,21,30,33]. In recent years, a few studies found that the time-varying parameter in the Budyko framework can better describe the changing processes of $E_a$ and separate the impacts of climate change and other land-related factors on the $E_a$ change in the changing environment, such as Jiang et al. [11], Liu et al. [39] and Tian et al. [40]. At the same time, combing the elasticity method and Budyko framework (such as the equation proposed by Turc-Pike et al. [41], Fu [35], Zhang et al., [36], Yang et al. [42] and Wang-Tang [37]), some studies specifically explored the impacts and contributions of climatic factors and land-related factors on the $E_a$ change (e.g., Yang and Yang [43], Jiang et al. [13], Xu et al. [44] and Wang et al. [45]). However, with the time-varying parameter in the Budyko framework, limited studies have specifically investigated the impacts and contributions of different numbers of climatic

factors and vegetation indices on $E_a$ change. Therefore, we try to establish the relationship between the time-varying parameter, n, and temporally changing variables and further quantify each related climatic factor and vegetation indices on $E_a$ change in this study.

In this study, based on the time window method, the temporal change in parameter n and the associated drivers are investigated to develop an empirical model of the time-varying parameter n by using a stepwise linear regression method. The motivation for advancing the empirical model of n is to use it in the context of three Budyko-type equations described by Fu [35], Zhang et al. [36] and Wang-Tang [37] to estimate $E_a$ and analyze the impacts of changes in climate and vegetation coverage on change in $E_a$. Specifically, the objectives of this paper are:

(1)  To explore the temporal trends of major meteorological factors and land surface vegetation coverage from 1982 to 2013 in the study area.
(2)  To improve $E_a$ estimations using the Budyko-type equations described by Fu [35], Zhang et al. [36] and Wang-Tang [37] by establishing the relationships between the time-varying parameter, n, and the indices of climate and vegetation coverage.
(3)  To analyze the impacts of climate change and vegetation variations on $E_a$ change by combining the improved Budyko-type equations and elasticity method.

The remainder of this paper is organized as follows. Descriptions of the datasets used in this study and the study area are introduced in Section 2. The methodology, including descriptions of the Budyko framework, moving window method, stepwise linear regression analysis, and contribution analysis, is given in Section 3. The results and discussions of the model assessment in relation to the datasets are given in Section 4. Finally, the conclusions are presented in Section 5.

## 2. Study Area and Data

The North and South Panjiang basin (NSPB) (102°12′ E–107°30′ E, 23°6′ N–26°48′ N), which is one of the Pearl River sub-basins (see Figure 1), was selected as the study catchment in this study. The drainage area of the NSPB is approximately 102,199 km², which occupies about 22.5% of the Pearl River basin. The climate of this catchment is a humid monsoon climate with an annual average precipitation of 1200 mm and an annual average temperature of 16.5 °C. The flood season begins in April and ends in September, and the precipitation during this period accounts for approximately 75% of the total annual precipitation. Table 1 presents the percentages of six major land use/cover types from 1980 to 2010 in the NSPB. It is found that the changes in each type of land use/cover from 1980 to 2010 are not obvious, and the rate of change of all types of land cover is less than 1%, thus, it can be assumed that the land use/cover in the NSPB are almost constant from 1980 to 2010.

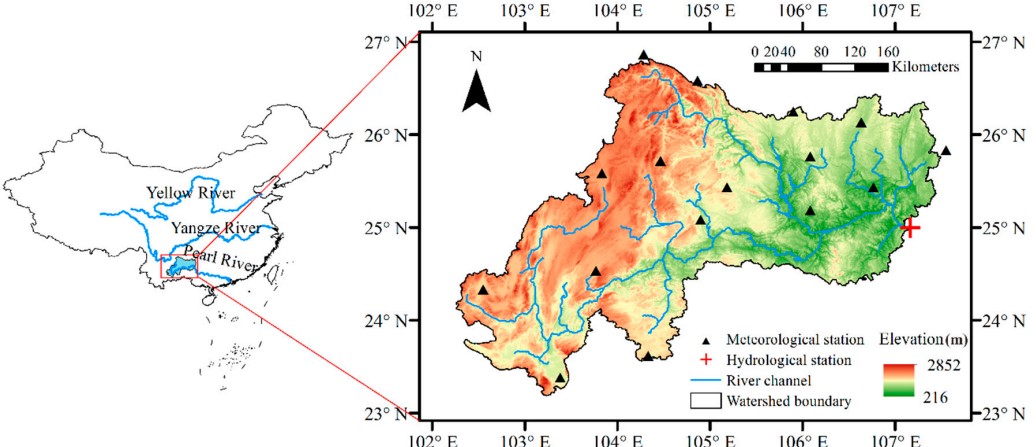

**Figure 1.** Map of the study area and locations of meteorological and hydrological stations used in this study.

**Table 1.** Percentages of six major land use/cover types in the North and South Panjiang basin (NSPB).

| Land Types | 1980s | 1990s | 1995s | 2000s | 2005s | 2010s |
|---|---|---|---|---|---|---|
| Agricultural land | 22.6% | 22.6% | 21.7% | 22.3% | 22.2% | 22.1% |
| Forest land | 50.7% | 50.5% | 51.0% | 50.3% | 50.6% | 50.5% |
| Grass land | 25.5% | 25.4% | 25.6% | 25.8% | 25.6% | 25.6% |
| Water area | 0.7% | 0.8% | 0.7% | 0.8% | 0.8% | 0.9% |
| Built-up land | 0.6% | 0.7% | 0.9% | 0.8% | 0.8% | 0.9% |
| Unused land | 0.1% | 0.1% | 0.1% | 0.1% | 0.1% | 0.1% |

Tian-e hydrologic station is the outlet of the NSPB, with an average runoff of about 450 mm during 1982 to 2013, and the annual runoff data from 1982 to 2013 at this outlet were obtained from the Ministry of Water Resources of China in this study. The meteorological data obtained from 1982 to 2013 at the 15 meteorological stations, which include daily precipitation (P), average temperature (T), maximum temperature ($T_{max}$), minimum temperature ($T_{min}$), sunshine duration (S), wind speed at a height of 10 m (u) and relative humidity (RH), were provided by the China Administration of Meteorology (http://www.data.cma.cn). Additionally, the value of potential evapotranspiration ($E_o$) was estimated using the FAO Penman–Monteith equation [46], which can estimate $E_o$ accurately under different climatic conditions [47,48]. The wind speed at a height of 2 m (noted as $u_2$) was calculated based on u using the profile relationship proposed by Allen et al. [46]. The net radiation ($R_n$) was calculated from S using the method recommended by Allen et al. [46], and the ground heat flux (G) was neglected because the daily and long-term average values of G are close to zero [46]. All the meteorological datasets noted above were spatially averaged by the Thiessen polygons method using ArcGIS software. To quantify the impacts of changes in vegetation coverage on $E_a$, the 8 km Global Inventory Modeling and Mapping Studies (GIMMS) Normalized Difference Vegetation Index (NDVI) dataset from 1982 to 2013 was derived from the Advanced Very High-Resolution Radiometer (AVHRR) sensor. Similar to climatic variables, the monthly NDVI dataset was used to represent the land surface vegetation conditions [49] of the catchment in this study. To analyze the process of land surface use/cover change in the NSPB from 1980 to 2010, a 1 km resolution land use/cover map was provided by the Resource and Environment Data Cloud Platform, Chinese Academy of Sciences (CAS) (http://www.resdc.cn/). In this study, the types of land use/cover are classified as forestland, grass land, agricultural land, built-up land, water area and unused land [21].

## 3. Methodology

### 3.1. Budyko Framework

The Budyko hypothesis indicates that terrestrial $E_a$ is subject to the common limitations of the water supply (usually measured by P) and energy supply (usually measured by $E_o$) [11]. According to this hypothesis, the water supply is the limiting factor of $E_a$ in arid catchment ($E_o/P > 1$), whereas energy availability is the limiting factor of $E_a$ in humid catchments ($E_o/P < 1$). Based on this approach, a coupling relationship between $E_a/P$ and $E_o/P$ was proposed, and the general form of this relationship can be expressed as follows [11]:

$$\frac{E_a}{P} = \left[ \frac{E_o}{P} \tanh(\frac{P}{E_o})[1 - \exp(-\frac{E_o}{P})] \right]^{0.5} \qquad (1)$$

where, the units of the variables in Equation (1) are mm. Thereafter, many similar equations have been developed [35–37]. In this study, the three Budyko-type equations proposed by Fu [35], Zhang et al. [36] and Wang-Tang [37] (see Table 2) were selected to investigate the impacts of climate change and vegetation variation on $E_a$ change. Compared with the original Budyko-type equations, which have no basin-specific parameter, the Budyko-type equations with basin-specific parameters are more flexible to account for the effects of different catchment characteristics on $E_a$ change [50]. In this study,

the basin-specific parameters in the three Budyko-type equations proposed by Fu [35], Zhang et al. [36] and Wang-Tang. [37] are expressed as $n^{Fu}$, $n^Z$ and $n^{W-T}$, respectively.

**Table 2.** The three Budyko-type equations considered in this study.

| Budyko-Type Equations | Reference |
|---|---|
| $\frac{E_a}{P} = 1 + \frac{E_o}{P} - [1 + (\frac{E_o}{P})^{n^{Fu}}]^{1/n^{Fu}}$ | Fu [35] |
| $\frac{E_a}{P} = (1 + n^Z \times \frac{E_o}{P})/(1 + n^Z \times \frac{E_o}{P} + \frac{P}{E_o})$ | Zhang et al. [36] |
| $\frac{E_a}{P} = (1 + \frac{E_o}{P} - \sqrt{(1 + \frac{E_o}{P})^2 - 4n^{W-T} \times (2 - n^{W-T}) \times \frac{E_o}{P}})/[4n^{W-T} - 2(n^{W-T})^2]$ | Wang-Tang [37] |

### 3.2. Moving Window Method

In general, the basic water balance in closed catchments can be expressed as follows:

$$E_a = P - R - \Delta S \tag{2}$$

where P is annual precipitation, R is annual runoff measured as streamflow and $\Delta S$ is the annual change of total water storage in the closed catchment. Because the Budyko-type equations can only be applied at the long-term average scale when $\Delta S$ is approximately zero (i.e., $\Delta S \approx 0$), the moving window method is applied to remove the influence of annual water storage changes in this study. Specifically, in the application of the moving window method, the width of the window size is related to the length of the variable series, and the longer window can better reduce the scatter of the variable series [13,51]; however, some specific evolution processes may be missed. According to previous studies [13,38,40], the changes of total water storage can be neglected during a long time period, and different sizes of time windows, such as 11 years and 13 years, were used in studies proposed by Jiang et al. [13] and Tian et al. [40], respectively. Here, as the P-R did not change much for time window size larger than 11 years, we selected the 11-year time window to minimize the influence of changes in $\Delta S$ series. In this study, the 11-year time window is applied to annual R, $E_a$, P and $\Delta S$ series, and their values in the time window centered at year t are expressed as $R_t$, $E_{at}$, $P_t$ and $\Delta S_t$, respectively. Thus, in the time window centered at year t, $\Delta S_t$, which mainly includes changes in ground water and the plant moisture content of the catchment, can be neglected (i.e., $\Delta St \approx 0$), and the water balance equation (i.e., Equation (2)) can be simplified as:

$$E_{at} = P_t - R_t. \tag{3}$$

Similar to R, $E_a$ and P, the 11-year time window is also used for other variables, including n, T, RH, $u_2$, $T_{min}$, $T_{max}$, S and NDVI, the values of which, in the time window centered at year t, are expressed as $n_t$, $T_t$, $RH_t$, $u_{2t}$, $T_{mint}$, $T_{maxt}$, $S_t$ and $NDVI_t$ in the following sections.

### 3.3. Modeling the Parameter $n_t$ in the Budyko-type Equations

For near-surface climate conditions, the basic meteorological factors, including $P_t$, $T_t$, $T_{maxt}$, $T_{mint}$, $RH_t$, $u_{2t}$ and $S_t$, were selected as indicators to describe the variability in $n_t$ in this study. For other factors related to land surface conditions, because the change in each type of land use/cover from 1980 to 2010 is not significant, and it can be assumed that the soil conditions and topography of a single catchment remain unchanged at the multiyear average scale, the natural variation in vegetation coverage described by NDVI in this study was selected as the only land surface variable other than climate to describe the variability in $n_t$ in the study area. In summary, eight candidate variables, namely, $P_t$, $T_t$, $T_{maxt}$, $T_{mint}$, $RH_t$, $u_{2t}$, $S_t$ and $NDVI_t$, were selected as the indicators of the catchment characteristics to investigate their respective relation to the variability in $n_t$ in each Budyko-type equation.

The stepwise linear regression method was used to determine the potential relationship between the parameter $n_t$ and the variables that were selected as the indicators of the catchment characteristics.

Moreover, an empirical model is considered to explain and formulize the potential relationships, and it can be expressed as follows:

$$n_t' = f(NDVI_t, P_t, T_t, RH_t, u_{2t}, S_t, T_{mint}, T_{maxt}), \tag{4}$$

where f is a function to be determined. The parameter $n_t$ obtained in this way is called the "modeled $n_t$", which is denoted as $n_t'$, and the $E_{at}$ value calculated by the Budyko-type equations with $n_t'$ is called "modeled $E_{at}$", which is denoted as $E_{at}'$. The performance of this empirical model can be evaluated by comparing the values of $n_t$ calculated by the Budyko-type equations using the observations to the values of $n_t'$. The resulting statistic, namely, the coefficient of determination ($r^2$) and root mean squared error (RMSE), were used to evaluate the performance of this empirical formula.

### 3.4. Contribution Analysis

According to Wang et al. [45], assuming that each single variable is independent in the Budyko-type equations (see Table 2), the elasticity method is used to analyze the contributions of climate and vegetation changes to $E_{at}$ change, and the first-order approximation of the changes in $E_{at}$ can be expressed as:

$$dE_{at} = \frac{\partial E_{at}}{\partial P_t}dP_t + \frac{\partial E_{at}}{\partial E_{ot}}dE_{ot} + \frac{\partial E_{at}}{\partial n_t}dn_t \tag{5}$$

$$
\begin{aligned}
\frac{dE_{at}}{E_{at}} &= \frac{\partial E_{at}}{\partial P_t}\frac{P_t}{E_{at}}\frac{dP_t}{P_t} + \frac{\partial E_{at}}{\partial E_{ot}}\frac{E_{ot}}{E_{at}}\frac{dE_{ot}}{E_{ot}} + \frac{\partial E_{at}}{\partial n_t}\frac{n_t}{E_{at}}\frac{dn_t}{n_t} \\
&= \varepsilon_P\frac{dP_t}{P_t} + \varepsilon_{E_{ot}}\frac{dE_{ot}}{E_{ot}} + \varepsilon_{n_t}\frac{dn_t}{n_t}
\end{aligned}
\tag{6}
$$

where $\varepsilon_{P_t}$, $\varepsilon_{E_{ot}}$ and $\varepsilon_{n_t}$ are the $P_t$, $E_{ot}$ and $n_t$ elasticities of $E_{at}$, and $\frac{dE_{ot}}{E_{ot}}$ can be written as follows based on the FAO Penman–Monteith equation [45]:

$$
\begin{aligned}
\frac{dE_{ot}}{E_{ot}} &= \frac{\partial E_{ot}}{\partial T_{maxt}}\frac{1}{E_{ot}}dT_{maxt} + \frac{\partial E_{ot}}{\partial T_{mint}}\frac{1}{E_{ot}}dT_{mint} + \frac{\partial E_{ot}}{\partial RH_t}\frac{RH_t}{E_{ot}}\frac{dRH_t}{RH_t} + \frac{\partial E_{ot}}{\partial S_t}\frac{S_t}{E_{ot}}\frac{dS_t}{S_t} + \frac{\partial E_{ot}}{\partial u_{2t}}\frac{u_{2t}}{E_{ot}}\frac{du_{2t}}{u_{2t}} \\
&= \varepsilon'_{T_{maxt}}dT_{maxt} + \varepsilon'_{T_{mint}}dT_{mint} + \varepsilon'_{RH_t}\frac{dRH_t}{RH_t} + \varepsilon'_{S_t}\frac{dS_t}{S_t} + \varepsilon'_{u_{2t}}\frac{du_{2t}}{u_{2t}}
\end{aligned}
\tag{7}
$$

where $\varepsilon'_{T_{maxt}}$, $\varepsilon'_{T_{mint}}$, $\varepsilon'_{RH_t}$, $\varepsilon'_{S_t}$ and $\varepsilon'_{u_{2t}}$ are the $T_{maxt}$, $T_{mint}$, $RH_t$, $S_t$ and $u_{2t}$ elasticities of $E_{ot}$. Substituting Equation (7) into Equation (6) yields the following expression:

$$
\begin{aligned}
\frac{dE_{at}}{E_{at}} &= \varepsilon_P\frac{dP_t}{P_t} + \varepsilon_{E_{ot}}\varepsilon'_{T_{maxt}}dT_{maxt} + \varepsilon_{E_{ot}}\varepsilon'_{T_{mint}}dT_{mint} + \varepsilon_{E_{ot}}\varepsilon'_{RH_t}\frac{dRH_t}{RH_t} + \varepsilon_{E_{ot}}\varepsilon'_{S_t}\frac{dS_t}{S_t} + \varepsilon_{E_{ot}}\varepsilon'_{u_{2t}}\frac{du_{2t}}{u_{2t}} + \varepsilon_{n_t}\frac{dn_t}{n_t} \\
&= \varepsilon_P\frac{dP_t}{P_t} + \varepsilon_{T_{maxt}}dT_{maxt} + \varepsilon_{T_{mint}}dT_{mint} + \varepsilon_{RH_t}\frac{dRH_t}{RH_t} + \varepsilon_{S_t}\frac{dS_t}{S_t} + \varepsilon_{u_{2t}}\frac{du_{2t}}{u_{2t}} + \varepsilon_{n_t}\frac{dn_t}{n_t}
\end{aligned}
\tag{8}
$$

where $\varepsilon_{T_{maxt}}$, $\varepsilon_{T_{mint}}$, $\varepsilon_{RH_t}$, $\varepsilon_{S_t}$ and $\varepsilon_{u_{2t}}$ are the $T_{maxt}$, $T_{mint}$, $RH_t$, $S_t$ and $u_{2t}$ elasticities of $E_{at}$. Thus, the changes in $E_{at}$ induced by variations in all the independent variables can be estimated as:

$$
\begin{aligned}
\Delta E_{at} &= \Delta E_{at-P_t} + \Delta E_{at-T_{maxt}} + \Delta E_{at-T_{mint}} + \Delta E_{at-RH_t} + \Delta E_{at-S_t} + \Delta E_{at-u_{2t}} + \Delta E_{at-n_t} \\
&= \varepsilon_P\frac{E_{at}}{P_t}\Delta P_t + \varepsilon_{T_{maxt}}E_{at}\Delta T_{maxt} + \varepsilon_{T_{mint}}E_{at}\Delta T_{mint} + \varepsilon_{RH_t}\frac{E_{at}}{RH_t}\Delta RH_t \\
&\quad + \varepsilon_{S_t}\frac{E_{at}}{S_t}\Delta S_t + \varepsilon_{u_{2t}}\frac{E_{at}}{u_{2t}}\Delta u_{2t} + \varepsilon_{n_t}\frac{E_{at}}{n_t}\Delta n_t
\end{aligned}
\tag{9}
$$

and the relative contributions of single variable to $E_{at}$ change can be calculated as follows:

$$\eta_{vi} = \frac{\Delta E_{at-vi}}{\Delta E_{at}} \times 100\% \tag{10}$$

where vi is a variable (such as $P_t$, $T_{maxt}$, $T_{mint}$, $RH_t$, $u_{2t}$, $S_t$), and $\eta_{vi}$ is the relative contribution of changes in vi to $E_{at}$ changes.

## 4. Results and Discussion

### 4.1. Temporal Variations in Climatic Variables and the NDVI

To investigate the temporal variations in meteorological factors and the NDVI from 1982 to 2013 in the NSPB, the nonparametric trend-free pre-whitening Mann–Kendall test developed by Mann [52], Kendall [53] and Yue et al. [54,55] was used in this section. As shown in Figure 2, significant warming can be observed in the study area, as the annual T, $T_{max}$ and $T_{min}$ values displayed an evident increasing trend at the 0.05 significance level, which is consistent with the general trends of the global warming [56] and warming in China [57]. Wind speed has significantly decreased, which supports the results given by McVicar et al. [58]. Similar to the wind speed, annual P and RH also exhibit significant decreasing trends. In addition, S presents a slight decreasing trend, and the NDVI in the study area displays a slight increasing trend (*p*-value > 0.05).

### 4.2. Modeling $n'_t$

A stepwise regression method was used to establish the potential relationship between the parameter $n_t$ and the eight candidate variables (discussed in Section 3.3) selected as the indicators of the catchment characteristics. The results of stepwise regression show that the parameter $n_t$ simulated with four candidate variables, including $S_t$, $NDVI_t$, $T_{maxt}$ and $P_t$, yields a satisfactory determination coefficient, $r^2$, between $n'_t$ and $n_t$ calculated by the Budyko-type equations using the observations. As shown in Table 3, the first candidate variable entered into the stepwise regression equation was $S_t$, which resulted in $r^2$ values greater than 0.5 among the three parameters, $n'_t$. The other three selected candidate variables, namely, $NDVI_t$, $T_{maxt}$ and $P_t$, increased $r^2$ by more than 0.02. The use of remaining candidate variables (such as $RH_t$, $u_{2t}$, $T_t$ and $T_{mint}$) did not effectively increase the $r^2$ value. Therefore, $S_t$, $NDVI_t$, $T_{maxt}$ and $P_t$ were identified as the decision variables and used to predict the parameter $n_t$ in this study, and the final fitted model of $n'_t$ in each Budyko-type equation was obtained as:

$$n_t^{Fu\prime} = -2.23 \ln S_t - 4.34 \ln NDVI_t + 9.77 \ln T_{maxt} + 0.82 \ln P_t - 27.55 \tag{11a}$$

$$n_t^{Z\prime} = -2.51 \ln S_t - 4.87 \ln NDVI_t + 11.24 \ln T_{maxt} + 0.58 \ln P_t - 33.00 \tag{11b}$$

$$n_t^{W-T\prime} = -0.56 \ln S_t - 1.05 \ln NDVI_t + 2.42 \ln T_{maxt} + 0.23 \ln P_t - 6.90 \tag{11c}$$

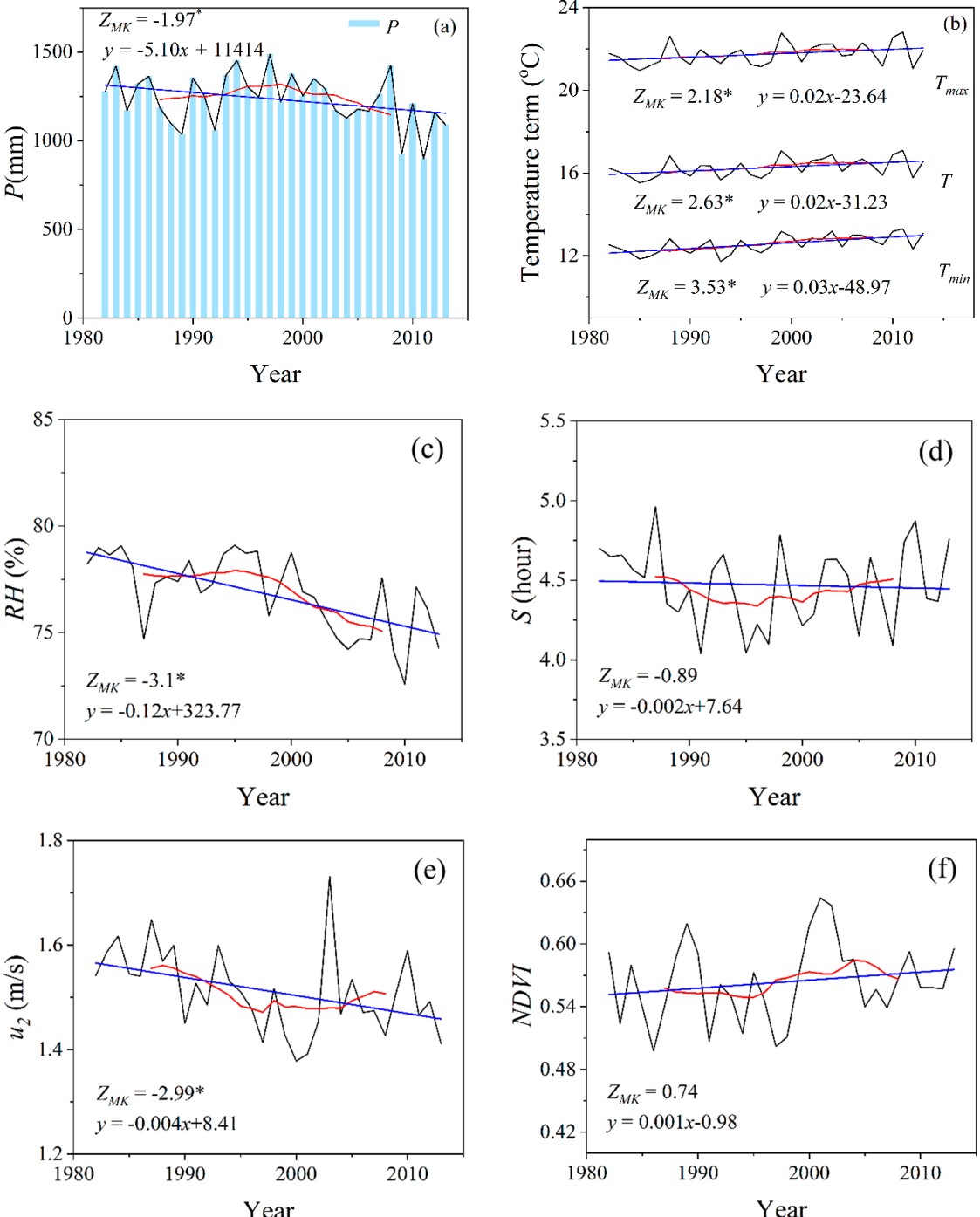

**Figure 2.** Changes in climatic variables and the Normalized Difference Vegetation Index (NDVI) in the NSPB. (**a**–**f**) are the changes in precipitation (P), temperature term, relative humidity (RH), sunshine duration (S), wind speed at a height of 2 m (u₂) and NDVI, respectively. The value of "$Z_{MK}$" is the trend of the variable over the period of 1982–2013 estimated using the nonparametric trend-free pre-whitening Mann–Kendall trend test. The symbol '*' after $Z_{MK}$ indicates that the estimated trend is significant at the 0.05 significance level (i.e., *p*-value < 0.05). The black lines represent the annual time series of variables, the blue lines represent the linear trends of the annual time series of variables and the red lines represent the 11-year moving averages of the annual time series of variables.

**Table 3.** Stepwise regression results between $n_t$ and the surface candidate variables in the NSPB.

| Variable | Step | $r^2$ | | |
|---|---|---|---|---|
| | | **Fu** | **Zhang** | **Wang-Tang** |
| $S_t$ | 1 | 0.62 | 0.57 | 0.65 |
| $S_t$, $NDVI_t$ | 2 | 0.71 | 0.64 | 0.72 |
| $S_t$, $NDVI_t$, $T_{maxt}$ | 3 | 0.79 | 0.76 | 0.79 |
| $S_t$, $NDVI_t$, $T_{maxt}$, $P_t$ | 4 | 0.82 | 0.78 | 0.84 |
| $S_t$, $NDVI_t$, $T_{amxt}$, $P_t$, $RH_t$, $u_{2t}$, $T_t$, $T_{mint}$ | 5 | 0.83 | 0.79 | 0.84 |

Figure 3 compares the values of $n'_t$ estimated using Equations (11a)–(11c), with the $n_t$ values estimated based on observations using each Budyko-type equation. The modeled $n'_t$ values are closely correlated with $n_t$, and the $r^2$ values are all greater than 0.78, suggesting that the relationships in these three empirical equations can accurately encompass the impacts of climatic factors and $NDVI_t$ on $n_t$. From these empirical formulas of modeled $n'_t$, it can be found that climate change affects $E_{at}$ by simultaneously influencing the meteorological inputs (such as $P_t$ and $E_{ot}$) and the basin-specific parameter $n_t$, while the $NDVI_t$ change affects $E_{at}$ implemented by the alteration of the parameter $n_t$ in the NSPB from 1982 to 2013. Moreover, the sign of each coefficient in Equations (11a)–(11c) can reflect the relationship between the changes in variables and $n'_t$ change. Generally, a positive (or negative) coefficient means that $n'_t$ will increase (or decrease) as the value of the variable increases. Thus, we can shortly draw from Equations (11a)–(11c) that $n'_t$ decreases as $S_t$ and $NDVI_t$ increase, and $n'_t$ increases as $P_t$ and $T_{maxt}$ increase. However, the sign of each coefficient in Equations (11a)–(11c) cannot reflect the contribution of each related change to $n'_t$ change, which will be further analyzed in the next section. Additionally, it is worth noting that the coefficient of each variable in Equations (11a)–(11c) is basin-specific, it should recalibrate in other catchments or regions which with different climate and vegetation conditions.

Additionally, the empirical formula for $n_t$ is used together with the Budyko-type equations to calibrate the $E_{at}$ estimation in the NSPB. In this study, two methods of $E_{at}$ estimation were considered for comparison: (i) using the time-varying $n'_t$ calculated by Equations (11a)–(11c) with time-varying variables and (ii) using the constant parameter $n_t$ calculated with the long-term average variables. Figure 4 shows the comparison between the values of $E_{at}$ obtained from the water balance equation (i.e., Equation (3)) and these two cases. The $r^2$ values with a constant $n_t$ vary from 0.55 to 0.61, and the $r^2$ values with a time-varying $n'_t$ are all larger than 0.90. This result indicates that the performance of each Budyko-type equation with a time-varying $n'_t$ is better than that for a constant $n_t$, which is consistent with the findings of Yang et al. [21], Jiang et al. [13] and Tian et al. [40]. Therefore, compared with the temporally constant parameter n which was used in most previous studies [25,32,33,44,59], the time-varying parameter n improves the accuracy of $E_a$ simulation based on the Budyko-type equations and specifically describes the evolution process of $E_a$ in the NSPB.

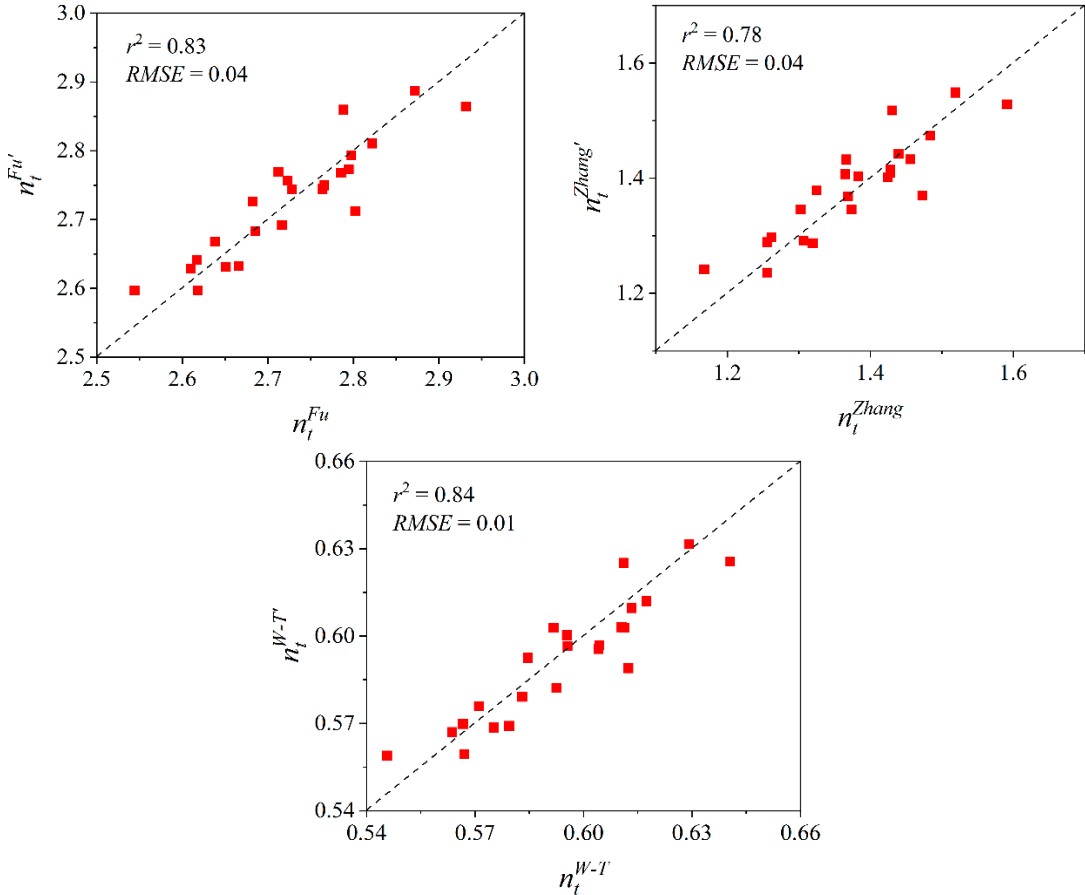

**Figure 3.** Comparison between the parameter $n'_t$ for each Budyko-type equation estimated by Equations (11a)–(11c) and $n_t$ derived from the Budyko-type equations using the observations. RMSE = root mean squared error.

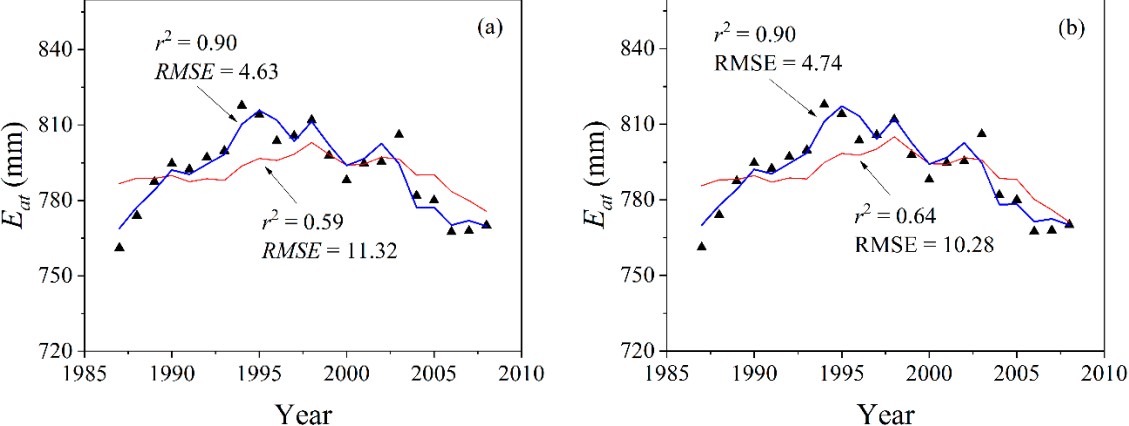

**Figure 4.** *Cont.*

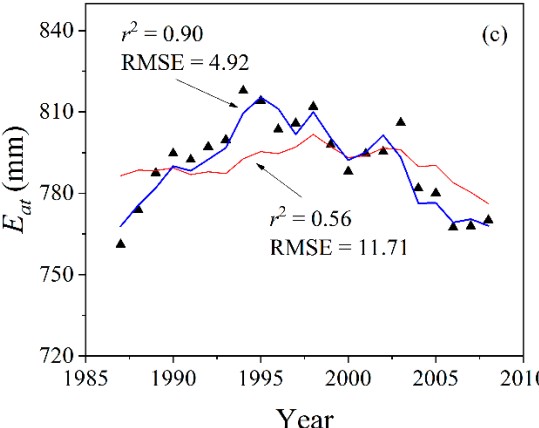

**Figure 4.** Comparison of $E_{at}$ calculated from the water balance equation (black triangles) and the simulated $E_{at}$ from the cases when the parameter $n_t$ in each Budyko-type equation is treated as a constant (red lines) and when the value of parameter $n_t$ is estimated using Equations (11a)–(11c) (blue lines). (**a**–**c**) are for the Budyko-type equations of Fu [35], Zhang et al. [36] and Wang-Tang [37], respectively.

## 4.3. Quantitative Contributions to $E_{at}$ Variation

Based on the analysis in Section 4.2, both climate change and vegetation variations will affect $E_{at}$ variations in the NSPB. Moreover, combining the empirical equation for $n_t$ estimation (i.e., Equations (11a)–(11c)) and the FAO Penman–Monteith equation, which was used to estimate $E_o$, the Budyko-type equations used in this study can be written as $E_{at} = f(P_t, T_{maxt}, T_{mint}, S_t, RH_t, u_{2t}, NDVI_t)$. Therefore, based on Equations (8) and (9), the change in $E_{at}$ ($\Delta E_{at}$) can be further expressed as:

$$
\begin{aligned}
\Delta E_{at} &= \Delta E_{at-P_t} + \Delta E_{at-T_{maxt}} + \Delta E_{at-T_{mint}} + \Delta E_{at-S_t} + \Delta E_{at-RH_t} + \Delta E_{at-u_{2t}} + \Delta E_{at-NDVI_t} \\
&= \varepsilon_{P_t} \frac{E_{at}}{P_t} \Delta P_t + \varepsilon_{T_{maxt}} E_{at} \Delta T_{maxt} + \varepsilon_{T_{mint}} E_{at} \Delta T_{mint} + \varepsilon_S \frac{E_{at}}{S_t} \Delta S_t + \varepsilon_{RH_t} \frac{E_{at}}{RH_t} \Delta RH_t + \varepsilon_{u_{2t}} \frac{E_{at}}{u_{2t}} \Delta u_{2t} \quad (12) \\
&\quad + \varepsilon_{NDVI_t} \frac{E_a}{NDVI_t} \Delta NDVI_t
\end{aligned}
$$

where $\Delta E_{at-P_t}$, $\Delta E_{at-T_{maxt}}$, $\Delta E_{at-T_{mint}}$, $\Delta E_{at-S_t}$, $\Delta E_{at-RH_t}$, $\Delta E_{at-u_{2t}}$ and $\Delta E_{at-NDVI_t}$ represent the changes in $E_{at}$ due to changes in $P_t$, $T_{maxt}$, $T_{mint}$, $S_t$, $RH_t$, $u_{2t}$ and $NDVI_t$, respectively. As in Equation (9), the elasticity of $E_{at}$ to climatic factors and $NDVI_t$ can be estimated as $\varepsilon_{P_t} = \frac{\partial f}{\partial P_t} \frac{P_t}{E_{at}}$, $\varepsilon_{T_{maxt}} = \frac{\partial f}{\partial T_{maxtt}} \frac{1}{E_{at}}$, $\varepsilon_{T_{mint}} = \frac{\partial f}{\partial T_{mint}} \frac{1}{E_{at}}$, $\varepsilon_{S_t} = \frac{\partial f}{\partial S_t} \frac{S_t}{E_{at}}$, $\varepsilon_{RH_t} = \frac{\partial f}{\partial RH_t} \frac{RH_t}{E_{at}}$, $\varepsilon_{u_{2t}} = \frac{\partial f}{\partial u_{2t}} \frac{u_{2t}}{E_{at}}$ and $\varepsilon_{NDVI_t} = \frac{\partial f}{\partial NDVI_t} \frac{NDVI_t}{E_{at}}$. In this section, the period of 1982–1992 is treated as the benchmark (pre-change period), and the following time widows are individually treated as post-change periods.

Equation (12) is used to analyze the contribution of climate change and vegetation variations to $E_{at}$ change ($\Delta E_{at}$) in each post-change period, which is estimated by subtracting the value of $E_{at}$ in the first time window (the period of 1982–1992). The results based on the equation proposed by Zhang et al. [36] were selected for analysis in this section, because the results of all three Budyko-type equations are very similar. First, the elasticity of $E_{at}$ to all variables in each time window is shown in Table 4. Notably, the elasticity of $E_{at}$ to $P_t$, $T_{maxt}$, $T_{mint}$ and $u_{2t}$ is positive, which indicates that an increase (decrease) in these variables will increase (decrease) $E_{at}$ in the study area. To the contrary, the elasticity of $E_{at}$ to $RH_t$, $S_t$ and $NDVI_t$ is negative, which indicates that an increase (decrease) in these variables will decrease (increase) $E_{at}$. In addition, the magnitude of the elasticity also reflects the sensitivity level of $E_{at}$ to these variables, the greater the absolute value of the elasticity is, the more impact the variable has on $E_{at}$. Thus, $E_{at}$ is more sensitive to $NDVI_t$ and $P_t$ than other variables in the NSPB, while the absolute values of elasticity are higher for $NDVI_t$ and $P_t$ than for other variables. Then, considering the changes of each variable in each time window relative to the values in the pre-change period (1982–1992) and the elasticity of $E_{at}$ to all variables, the contributions of $P_t$, $T_{maxt}$, $T_{mint}$, $RH_t$, $S_t$, $u_{2t}$ and $NDVI_t$ to $\Delta E_{at}$ can be obtained, and the results are shown in Table 5. From this table, it is found

that the changes in $P_t$, $T_{maxt}$ and $NDVI_t$ contributed most to $\Delta E_{at}$, while the absolute values of these contributions are relatively high compared with those of other factors. The changes in $u_{2t}$ and $T_{mint}$ have limited impacts on $\Delta E_{at}$ because the corresponding absolute values of the contributions are very low. Moreover, $T_{maxt}$ and $T_{mint}$ positively contribute to the $E_{at}$ increase in all time windows, because the values of elasticity and changes are all positive during the study period. $S_t$ makes a positive contribution to the $E_{at}$ increase because the values of elasticity and changes are all negative during the study period. Because the elasticity of $E_{at}$ to $P_t$ is positive in all time windows and $P_t$ increased in time windows from 1983–1993 to 1998–2008 but decreased during the time windows from 1999–2009 to 2003–2013, $P_t$ contributed positively to the $E_{at}$ increase during the time windows from 1983–1993 to 1998–2008 but contributed negatively during the time windows from 1999–2009 to 2003–2013. Similarly, the changes in $NDVI_t$, $RH_t$ and $u_{2t}$ all showed both positive and negative contributions to the $E_{at}$ increase. Additionally, the contribution of climate change ($\eta_C$) to $\Delta E_{at}$ can be obtained by summing all the contribution rates of related climatic variables (i.e., $\eta_C = \eta_{P_t} + \eta_{T_{maxt}} + \eta_{T_{mint}} + \eta_{RH_t} + \eta_{S_t} + \eta_{u_{2t}}$), while the contribution of vegetation change ($\eta_V$) to $\Delta E_{at}$ is obtained by the contribution rate of NDVI (i.e., $\eta_V = \eta_{NDVI_t}$). Overall, climate change contributed more to $\Delta E_{at}$ than vegetation change, and the change in vegetation contributed positively during the time windows from 1982–1992 to 1992–2002 but negatively during the time windows from 1993–2003 to 2003–2013.

**Table 4.** Elasticity of $E_{at}$ to variables in different time windows based on the equation proposed by Zhang et al. [36].

| Time Window | Elasticity of $E_{at}$ to Variables | | | | | | |
|---|---|---|---|---|---|---|---|
| | $\varepsilon_{P_t}$ | $\varepsilon_{T_{maxt}}$ | $\varepsilon_{T_{mint}}$ | $\varepsilon_{RH_t}$ | $\varepsilon_{S_t}$ | $\varepsilon_{u_{2t}}$ | $\varepsilon_{NDVI_t}$ |
| 1982–1992 | 0.56 | 0.09 | 0.006 | −0.36 | −0.22 | 0.03 | −0.75 |
| 1983–1993 | 0.55 | 0.09 | 0.006 | −0.36 | −0.21 | 0.03 | −0.73 |
| 1984–1994 | 0.55 | 0.08 | 0.006 | −0.35 | −0.19 | 0.03 | −0.70 |
| 1985–1995 | 0.54 | 0.08 | 0.006 | −0.35 | −0.18 | 0.03 | −0.68 |
| 1986–1996 | 0.54 | 0.08 | 0.006 | −0.35 | −0.18 | 0.03 | −0.67 |
| 1987–1997 | 0.54 | 0.08 | 0.006 | −0.35 | −0.18 | 0.03 | −0.67 |
| 1988–1998 | 0.54 | 0.08 | 0.006 | −0.35 | −0.17 | 0.03 | −0.65 |
| 1989–1999 | 0.53 | 0.08 | 0.006 | −0.35 | −0.16 | 0.03 | −0.64 |
| 1990–2000 | 0.53 | 0.08 | 0.006 | −0.35 | −0.16 | 0.03 | −0.64 |
| 1991–2001 | 0.53 | 0.08 | 0.006 | −0.36 | −0.17 | 0.03 | −0.66 |
| 1992–2002 | 0.53 | 0.08 | 0.006 | −0.35 | −0.18 | 0.03 | −0.69 |
| 1993–2003 | 0.53 | 0.08 | 0.006 | −0.36 | −0.18 | 0.04 | −0.68 |
| 1994–2004 | 0.54 | 0.08 | 0.006 | −0.35 | −0.19 | 0.04 | −0.70 |
| 1995–2005 | 0.55 | 0.08 | 0.006 | −0.34 | −0.19 | 0.04 | −0.70 |
| 1996–2006 | 0.55 | 0.08 | 0.006 | −0.33 | −0.19 | 0.04 | −0.68 |
| 1997–2007 | 0.56 | 0.08 | 0.006 | −0.33 | −0.18 | 0.04 | −0.67 |
| 1998–2008 | 0.55 | 0.08 | 0.005 | −0.32 | −0.19 | 0.04 | −0.68 |
| 1999–2009 | 0.56 | 0.08 | 0.005 | −0.32 | −0.21 | 0.04 | −0.72 |
| 2000–2010 | 0.57 | 0.08 | 0.005 | −0.31 | −0.21 | 0.04 | −0.71 |
| 2001–2011 | 0.58 | 0.08 | 0.005 | −0.31 | −0.20 | 0.04 | −0.70 |
| 2002–2012 | 0.59 | 0.08 | 0.005 | −0.30 | −0.19 | 0.04 | −0.67 |
| 2003–2013 | 0.59 | 0.08 | 0.005 | −0.29 | −0.18 | 0.04 | −0.64 |

**Table 5.** The contributions of variables in different time windows based on the equation proposed by Zhang et al. [36] and $\Delta E_{a\ t}$ for each time window was estimated by Equation (12).

| Time Window | $\Delta E_{at}$ (mm) | Contribution Rate of Variables to $\Delta E_{a\ t}$ (%) | | | | | | | |
|---|---|---|---|---|---|---|---|---|---|
| | | $\eta_{P_t}$ | $\eta_{T_{maxt}}$ | $\eta_{T_{mint}}$ | $\eta_{RH_t}$ | $\eta_{S_t}$ | $\eta_{u_{2t}}$ | $\eta_C$ | $\eta_V=\eta_{NDVI_t}$ |
| 1982–1992 | 0 | 0 | 0 | 0 | 0 | 0 | 0 | 0 | 0 |
| 1983–1993 | 12.83 | 34.97 | 11.90 | −1.46 | 3.82 | 1.44 | 1.05 | 51.71 | 48.29 |
| 1984–1994 | 26.34 | 25.17 | 35.30 | 0.74 | 2.70 | 5.70 | 0.01 | 69.62 | 30.38 |
| 1985–1995 | 33.54 | 34.11 | 30.23 | 1.36 | 1.17 | 11.47 | −0.71 | 77.63 | 22.37 |
| 1986–1996 | 31.30 | 26.57 | 31.57 | 1.85 | 1.81 | 17.35 | −1.28 | 77.88 | 22.12 |
| 1987–1997 | 35.97 | 36.89 | 26.22 | 1.98 | 0.57 | 19.26 | −1.87 | 83.04 | 16.96 |
| 1988–1998 | 38.61 | 35.25 | 24.96 | 2.21 | −0.70 | 17.67 | −2.32 | 77.07 | 22.93 |
| 1989–1999 | 56.70 | 43.88 | 25.59 | 2.16 | −0.46 | 11.46 | −2.20 | 80.43 | 19.57 |
| 1990–2000 | 52.97 | 50.34 | 23.71 | 2.15 | −1.17 | 10.29 | −2.68 | 82.63 | 17.37 |
| 1991–2001 | 42.48 | 53.41 | 26.77 | 2.66 | −0.90 | 12.37 | −3.11 | 91.19 | 8.81 |
| 1992–2002 | 44.65 | 65.39 | 42.84 | 3.17 | 0.40 | 11.29 | −4.06 | 119.03 | −19.03 |
| 1993–2003 | 50.80 | 61.11 | 47.32 | 3.99 | 1.21 | 8.74 | −2.53 | 119.86 | −19.86 |
| 1994–2004 | 36.74 | 58.43 | 57.25 | 5.28 | 3.64 | 12.30 | −3.80 | 133.1 | −33.10 |
| 1995–2005 | 26.95 | 49.46 | 70.47 | 7.16 | 9.64 | 19.38 | −5.11 | 151.01 | −51.01 |
| 1996–2006 | 33.51 | 32.76 | 82.19 | 7.23 | 12.91 | 11.13 | −4.99 | 141.23 | −41.23 |
| 1997–2007 | 34.21 | 29.32 | 79.32 | 6.75 | 14.22 | 7.70 | −4.39 | 132.92 | −32.92 |
| 1998–2008 | 44.92 | 30.14 | 98.67 | 8.61 | 19.26 | 10.48 | −5.52 | 161.64 | −61.64 |
| 1999–2009 | 20.77 | −3.94 | 224.44 | 19.62 | 47.74 | 27.45 | −12.82 | 302.48 | −202.48 |
| 2000–2010 | 18.91 | −52.85 | 275.52 | 22.48 | 64.00 | 15.93 | −11.57 | 313.52 | −213.52 |
| 2001–2011 | 6.34 | −187.26 | 413.65 | 35.37 | 108.67 | 17.64 | −16.13 | 371.94 | −271.94 |
| 2002–2012 | 6.78 | −352.40 | 470.48 | 43.69 | 133.65 | 15.94 | −15.96 | 295.39 | −195.39 |
| 2003–2013 | 8.99 | −317.01 | 380.31 | 36.46 | 114.56 | 7.11 | −14.01 | 207.42 | −107.42 |

The contributions to $E_{at}$ variation obtained in this study matched those estimated in previous studies in China, such as by Yang and Yang [43] and Wang et al. [45], who demonstrated that precipitation changes were mainly responsible for the variations of evapotranspiration and runoff among the basic climatic variables. Additionally, for the temperature index, the elasticity of $E_{at}$ to $T_{maxt}$ is significantly greater than that to $T_{mint}$ in the NSPB, mainly because $T_{max}$ was used to calculate several important terms in the $E_o$ and $E_a$ estimation process, such as net radiation at the crop surface ($R_n$), the saturation vapor pressure ($e_s$), and the slops of the saturation vapor pressure versus air temperature curve ($\Delta$). Additionally, the values of $T_{max}$ are always greater than $T_{mint}$ at any time of the day and in any season [60]. Furthermore, the variation in the NDVI, which was selected as the indicator of vegetation conditions, has a large influence on the change in $E_a$. In fact, many studies have explored how vegetation coverage influences basin-scale water and energy balances based on the Budyko framework. For instance, Zhang et al. [36] suggested that a vegetation type-specific parameter can better express the water and energy balances in different catchments. Yang et al. [21] found that the time varying NDVI can be used to capture the variability of parameter n in the small catchments (<50,000 km$^2$) of northern China. Li et al. [22] established the relationship between the NDVI and the parameter in Fu's equation in 26 global river basins (>300,000 km$^2$). In addition, Abatzoglou and Ficklin [30] and Xing et al. [33] also used the NDVI to model n in the Budyko-type equations. The relationship between $n_t$ and $NDVI_t$ obtained in this study displays good agreement with the relationships obtained in previous studies, and it can be used to quantitatively assess the contribution of $NDVI_t$ variations to changes in $E_{at}$.

### 4.4. Uncertainty in the Contribution Estimation

It is worth noting that adding more variables to the elasticity and contribution estimations may generate more errors and increase uncertainty [45]. Figure 5 describes the comparison between the simulated $\Delta E_{at}$ estimated by Equation (12) and the observed $\Delta E_{at}$ in each post-change time window. Although a good correlation ($r^2 \geq 0.85$) was obtained between the simulated $\Delta E_{at}$ and observed values,

a considerable RMSE existed in the contribution analysis. First, the error of the modeled $\Delta E_{at}$ was associated with the empirical model for parameter $n_t$ estimation, because the empirical models of $n_t$ established in this study cannot fully simulate the observed values and therefore introduce error into the $\Delta E_{at}$ simulation. Second, the approximation of the first-order Taylor expansion in Equation (12), which ignores the second-order and higher terms of the Taylor expansion [61], may be one of the reasons for the error between the modeled $\Delta E_{at}$ and the observed $\Delta E_{at}$. Additionally, there are other causes of the error between the observed $E_{at}$ change and the modeled $E_{at}$ change. For example, the interactions between input elements could introduce uncertainty into the attribution analysis, while some of them are not fully independent [24,33]. The temporal and spatial variability in the empirical parameters used for $E_o$ estimation may increase the uncertainty in the attribution analysis [62]. Therefore, the errors and uncertainties that exist in the elasticity and contribution analyses must be further studied in the future.

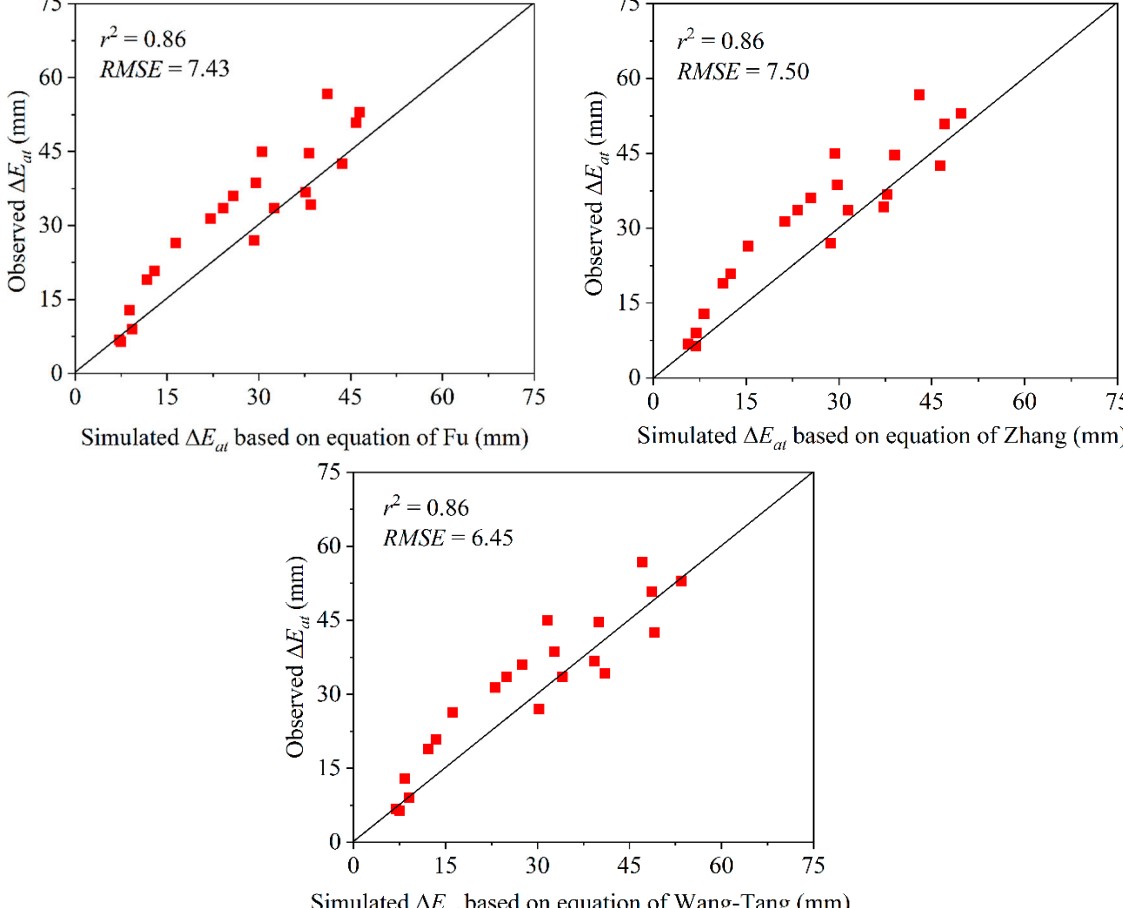

**Figure 5.** Comparison of simulate $\Delta E_{at}$ with Equation (12) and the observed values in each post-change time window.

## 5. Conclusions

In this study, an empirical model of the time-varying parameter $n_t$ was established to quantify the impacts of climate change and vegetation coverage on actual evapotranspiration. $E_{at}$ in the NSPB of China was simulated, and the contribution of each climatic variable and the NDVI to $E_{at}$ change was assessed based on the three Budyko-type equations proposed by Fu [35], Zhang et al. [36] and Wang-Tang [37]. The conclusions are drawn as follows.

(1) Significant warming was observed in the NSPB, and P, $u_2$ and RH significantly decreased. For land surface conditions, the land use/cover in this catchment changed slightly from 1980 to 2010,

and the NDVI, which was selected as the only land surface variable to describe the variability in parameter $n_t$ in the study area, displayed a slight increasing trend.

(2)    The stepwise linear regression method showed that combining three basic climatic factors and $NDVI_t$ can well explain the changes in the basin-specific $n_t$ in the Budyko-type equations, with $r^2$ values between the modeled $n'_t$ and $n_t$ calculated based on the Budyko-type equations using observations all being greater than 0.78. This finding indicated that the time-varying $n_t$ proposed in this study yielded better performance of the modeling $E_{at}$ than a constant $n_t$ in the study area.

(3)    Based on the elasticity and contribution analyses of $E_{at}$, it was found that $P_t$, $NDVI_t$ and $T_{maxt}$ are the major driving forces to the $E_{at}$ variations in the study area, and they contributed the most to the $E_{at}$ variations. During the study period, climate change contributed (whose average contribution is 149.6%) more to $\Delta E_{at}$ than vegetation change (whose contribution rate is −49.4%). Furthermore, the results also indicate that in addition to climatic factors, considering the impacts of vegetation coverage variation on actual evapotranspiration and the water cycle is very important in the study area.

**Author Contributions:** Conceptualization: T.L., J.X. and D.S.; data collection and analysis: T.L., L.C., L.Z. and B.L.; drafting the manuscript: T.L. and D.S.; review and editing: T.L. and D.S. All authors have read and agreed to the published version of the manuscript.

**Funding:** This work was supported by the National Natural Science Foundation of China (41890823).

**Acknowledgments:** The authors would like to thank Jun Xia, Dunxian She and Lei Cheng from the Wuhan University for their valuable suggestion and guidance on this article.

**Conflicts of Interest:** The authors declare no conflict of interest.

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
