# Peer review of "Quantifying the Impacts of Climate Change and Vegetation Variation on Actual Evapotranspiration Based on the Budyko Hypothesis in North and South Panjiang Basin, China"

_water, doi:10.3390/w12020508_

Round 1
Reviewer 1 Report
The article is interesting, and the topic is current. However, it needs some minor revisions.
In the Abstract, the acronym NDVI and the parameter n must be written in full
1.Introduction
Line 95 – explain why was selected the period 1982 to 2013
Study Area and DataLine 122 – indicate which area is dominated by the hydrological station. Indicate the average runoff. Are there no other stations in the study area?
Line 130 - parameter u – must be written in full
Line 140 – I don’t understand “1-km”
MethodologyLine 152 – indicate the units of the variables
Lines 159-160 –indicate the meaning of the exponents: Fu, Z, W-T
Author Response
Response to Reviewer 1 Comments
Point 1: In the Abstract, the acronym NDVI and the parameter n must be written in full.
Response 1: Thank for the concern. The NDVI represents the Normalized Difference Vegetation Index, and the parameter n in the Budyko-type equations is the single parameter, which usually used to describe the land surface characteristics. We have provided the full name of NDVI and the parameter n in the abstract (Page 1, Line 22-23.).
Point 2: Line 95 – explain why was selected the period 1982 to 2013.
Response 2: Thanks for the concern. Multiple source of data are used in this study, and different data sets has different time range. For example, the meteorological data (precipitation, maximum temperature, average temperature, minimum temperature, sunshine duration, wind speed and relative humidity) covers the time period of 1980-2015, runoff data in Tian-e station covers the time period of 1980-2013, and the NDVI data derived from the Advanced Very High Resolution Radiometer (AVHRR) sensor covers the period of 1982-2013. To balance the different ranges of various data-sets, the final time period is selected as 1982 to 2013. We confirm that these data-sets are sufficient for the analysis of impacts of climate change and vegetation variation on actual evaporation change in this study (see the section of results).
Point 3: Line 122 – indicate which area is dominated by the hydrological station. Indicate the average runoff. Are there no other stations in the study area?
Response 3: Thanks for the concern. The chosen hydrological station, Tian-e station, is the outlet of the study area, it can dominate the whole research region. The average runoff for the study area is about 450 mm. There are also some other hydrological stations in the NSPB, however, the main concern of our studies is the total runoff in the study area, therefore, only the Tian-e station has been used. We have provided the average runoff of Tian-e station in the revised manuscript (Page 4, Line 129.).
Point 4: Line 130 – parameter u – must be written in full.
Response 4: Thanks for the concern. The meaning of “u” is the wind speed at a height of 10 m above the ground, which has been mentioned in line 133-134.
Point 5: Line 140 – I don’t understand “1-km”.
Response 5: Thanks for the concern. The meaning of the “1-km” is that the map resolution of the land use/cover for the study area is 1 km, we revised the expression as “1 km resolution” in the revised manuscript (Page 4, Line 147).
Point 6: Line 152 – Indicate the units of the variables.
Response 6: Thanks for the concern. Usually, the units of the variables (Ea, P, Eo) in equation (1) are mm. We provided the units in the revised manuscript (Page 4, Line 160.).
Point 7: Line 159-160 – Indicate the meaning of the exponents: Fu, Z, W-T.
Response 7: Thanks for the concern. The exponents, Fu, Z, W-T, mean the corresponding method, which are equations proposed by Fu [1], Zhang [2] and Wang-Tang [3], respectively. In this study, three Budyko-type equations proposed by Fu [1], Zhang et al. [2] and Wang-Tang [3] were selected to investigate the impacts of climate change and vegetation coverage on actual evapotranspiration. Each of these three Budyko-type equations has a single basin-specific parameter, which expressed as nFu, nZ and nW-T, respectively. Therefore, the meaning of the exponents Fu, Z and W-T in nFu, nZ and nW-T represent the single parameter in Budyko-type equations proposed by Fu [1], Zhang et al. [2] and Wang-Tang [3], respectively.
References:
Fu, B.P. The calculation of the evaporation from land surface. Scientia Atmospherica Sinica. 1981, 5, 23-31. (In Chinese) Zhang L.; Dawes, W.R.; Walker, G.R. Response of mean annual evapotranspiration to vegetation changes at catchment scale. Water Resour. Res. 2001, 37, 701. Wang, D.; Tang, Y. A one-parameter Budyko model for water balance captures emergent behavior in darwinian hydrologic models. Res. Lett. 2014, 41, 4569–4577.

Reviewer 2 Report
Please see the attached file.

Author Response
Response to Reviewer 2 Comments
Point 1: Line 1: No quantification was made.
Response 1: Thanks for the concern. In fact, we have quantified the contribution rate of each related climatic variable (Pt, Tmaxt, Tmint, RHt, St and u2t) and NDVIt to the actual evapotranspiration in this study, and the results were shown in Table 5. The quantitative impacts of climate change and vegetation change were obtained based on the contribution rate of these single variables. Concretely, the contribution of climate change to the actual evapotranspiration change can be obtained by summing all the contribution rate of related climatic variables, while the contribution of vegetation change to the actual evapotranspiration change obtained by the contribution rate of NDVI. We added quantitative analysis in section 4.3 (Page 11, 347-353.).
Point 2: Line 3: Actual evapotranspiration means sum of evaporation and transpiration; however, in hydrology and here, you only focused on Ea which is evaporation, it is better to clarify the vague point all over the paper.
Response 2: Thanks for the concern. In general, actual evapotranspiration parameterized as a sum of soil evaporation, vegetation evaporation, and vegetation transpiration (Wang and Dickinson, 2012). The basis water balance equation used in this study is: , where P is total precipitation, R is the runoff in the outlet of the catchment, is the total changes of water storage, therefore, the Ea in this equation, which is also the core element of this study, represents the total evapotranspiration (the sum of soil evaporation, vegetation evaporation and vegetation transpiration) for the study area in this study. Thus, we selected “actual evapotranspiration” in this study.
References:
Wang, K.; Dickinson, R.E. a Review of Global Terrestrial Evapotranspiration : Observation, Modeling, Climatology, and climatic Variability. Rev. Geophys. 2012, 50, 1–54.
Point 3: Line 14: Better to use ET instead of Ea.
Response 3: Thanks for the concern. In this study, we used "Ea" to represent actual evapotranspiration, where ‘a’ means actual. The reason of using “Ea” is to distinguish the expression of actual evapotranspiration (Ea) and potential evapotranspiration (Eo).
Point 4: Line 18: Replace with another verb, improved is too strong.
Response 4: Thanks for the concern. We changed “improved” to “modified” in the revised manuscript (Page 1, Line 26.).
Point 5: Line 19-21: Need to rewrite and make it more clear. Bring the objective first. Also, 11 year time window data for what? And for where?
Response 5: Thanks for the suggestion. In order to remove the influence of the change of total water storage in the storage, the 11-year time window was used for variables (P, R, S, T, Tmax, Tmin, S, u2, RH and NDVI) which considered in this study. Based on the advice, we revised the expression as “First, in order to establish the relationship between the parameter n, which usually presents the land surface characteristics in the Budyko-type equations, with basic climatic variables and the Normalized Difference Vegetation Index (NDVI), the stepwise linear regression method has been used.” in the revised manuscript (Page 1, Line 21-24.).
Point 6: Line 26: Need add “for”.
Response 6: Thanks for the concern. We added “for” in the revised manuscript (Page 1, Line 29.).
Point 7: Line 29-32: This finding is so weak, also, concluding the contribution of parameters on evapotranspiration n is so obvious and it would be better to provide more statistical analysis than general word in the abstract.
Response 7: Thanks for the suggestion. Based on the quantifying of the impacts of climate change and vegetation variation on Eat change (Page 11, Line 346-352 and Table 5.), we have revised the expression as “The results of the elasticity and contribution analyses suggest that climate change contributed (whose average contribution is 149.6%) more to Ea change than vegetation variation (whose average contribution is -49.4%). Precipitation, NDVI and the maximum temperature are the major drivers of Ea change, while the minimum temperature and wind speed contribute the least to Ea change.” in the revised manuscript (Page 1, Line 29-33.).
Point 8: Line 37-39: The sentence is wrong! ET is sum of evaporation and transpiration not only evaporation, check reference i.e.: https://doi.org/10.3390/w11122478
Response 8: Thanks for the concern. Based on the studies proposed by Niaghi and Jia (2019), we changed “Driven mainly by solar radiation, terrestrial evapotranspiration (Ea) connects the water exchange between the land surface and atmosphere and balances the heat between the land surface and atmosphere through the transformation of water from liquid (in snow, ice or the soil) to gaseous form.” to “Driven mainly by solar radiation, terrestrial actual evaporation (Ea) plays a key role in the water and energy exchange between land surface and atmosphere.” in the revised manuscript (Page 1, Line 38-39.).
References:
Niaghi, A.R.; Jia, X. New approach to improve the soil water balance method for evapotranspiration estimation. Water (Switzerland) 2019, 11, 1–16.
Point 9: Line 46: Better to include the real studies on evaporation and also evapotranspiration as mentioned before.
Response 9: Thanks for the suggestion. We added the studies proposed by Douville et al. (2013) and McVicar et al. (2012) (Page 2, Line 45.). McVicar et al. found that evaporation is principally driven by radiative and aerodynamic processes and that controlling aerodynamic factors is mainly affected by the wind speed and atmospheric humidity. Douville et al. indicated that global warming have altered global evaporation in recent years.
References:
Douville, H.; Ribes, A.; Decharme, B.; Alkama, R.; Sheffield, J. Anthropogenic influence on multidecadal changes in reconstructed global evapotranspiration. Nat. Clim. Chang. 2013, 3, 59–62.
McVicar, T.R., Roderick, M.L., Donohue, R.J., Niel, T.G.V. Ecohydrology bearings- invited commentary-less bluster ahead? Ecohydrological implications of global trends of terrestrial near-surface wind speeds. Ecohydrology 2012, 5, 381–388.
Point 10: Line 46-47: Main driving factors can be lots of things, for example, in agriculture, you can find it: https://doi.org/10.1016/j.agwat.2019.105760. Make the sentence clear.
Response 10: Thanks for the concern. Based on the studies proposed by Niaghi et al. (2019), we changed “These findings have increase in the interest in investigating the potential driving factors that control the changes in Ea.” to “These findings have increased the interest in investigating the potential climatic driving factors that control the changes in Ea” in the revised manuscript (Page 2, Line 46.).
References:
Niaghi, A.R.; Jia, X. New approach to improve the soil water balance method for evapotranspiration estimation. Water (Switzerland) 2019, 11, 1–16.
Point 11: Line 63-66: You have to indicate agricultural sector when it comes to evapotranspiration related studies.
Response 11: Thanks for the suggestion. Undoubtedly, agricultural activities can make a significant impacts on the changes in Ea. For example, Jaramillo et al. (2013) found that a main increase of Ea is mainly due to the increase of cultivated area and/or crop production in Sweden. Therefore, we introduced the studies proposed by Jaramillo et al. (2013) to indicate the impacts of agriculture on the evaporation changes in the revised manuscript (Page 2, Line 64-65.).
References:
Jaramillo, F.; Prieto, C.; Lyon, S.W.; Destouni, G. Multimethod assessment of evapotranspiration shifts due to non-irrigated agricultural development in Sweden. J. Hydrol. 2013, 484, 55–62.
Point 12: Line 67: This is the main source of evapotranspiration! The language of writing need to be modified.
Response 12: Thanks for the suggestion. We changed “Some studies have suggested that the incorporation of key index of vegetation conditions into the water and energy balance of catchments is necessary” to “Previous studies have suggested that it is necessary to explicitly incorporate key indices of vegetation conditions into the Budyko framework for many catchments” in the revised manuscript (Page 2, Line 66-68.).
Point 13: Line 109-110: Occupying 22.5% of the Pearl River basin, the drainage area of the NSPB is approximately 102199 km2.
Response 13: Thanks for the suggestion. The NSPB that selected as the study area in this study is the sub-basin of the Pearl River basin, and its drainage occupying 22.5% of the total Pearl River basin. We revised “Occupying 22.5% of the Pearl River basin, the drainage area of the NSPB is approximately 102199 km2” to “The drainage area of the NSPB is approximately 102199 km2, which occupies about 22.5% of the Pear River basin” in the revised manuscript (Page 3, Line 117-118.).
Point 14: Line 111: The flooding season.
Response 14: I’m sorry for the mistake. We changed “flooding season” to “flood season” in the revised manuscript (Page 3, Line 119.).
Point 15: Data should be kept to one decimal place in Table 1.
Response 15: According to your advice, we have corrected the data in Table 1 in the revised manuscript.
Point 16: Line 129-131: clarification required.
Response 16: Thanks for the concern. As mentioned in Line 133-134, “u” in Line 138 represents wind speed at a height of 10 (m) above the ground, “S” in Line 139 represents sunshine duration.
Point 17: Line 146: Since you defined it, you have to use abbreviation.
Response 17: Thanks for the suggestion. We changed “actual evapotranspiration” to “Ea” in the revised manuscript (Page 4, Line 153.).
Point 18: Line 147-148: Is Eo indicate the energy supply? Or potential evapotranspiration rate?
Response 18: Thanks for the concern. Eo means the potential evapotranspiration in this study. Base on the studies proposed by Yang et al. (2009), Eo is used to measure the availability of energy, while P is used to measure the availability of water in the Budyko framework.
References:
Yang, D.; Shao, W.; Yeh, P.J.F.; Yang, H.; Kanae, S.; Oki, T. Impact of vegetation coverage on regional water balance in the nonhumid regions of China. Water Resour. Res. 2009, 45, 1–13.
Point 19: Line 163: “simplified”.
Response 19: Thanks for the suggestion. We removed “simplified” in the revised manuscript (Page 5, 170.).
Point 20: Line 163: Are you sure delta S is the annual change of water storage? How do you measure the annual change?
Response 20: Thanks for the concern. The DS defined in this study means the change of total stored water in the catchments. In fact, it is not necessary to measure the annual change of water storage in this study, because the DS can be neglected if the study period is long enough, e.g. decades, in the closed catchment. Therefore, we changed “DS is the annual change of water storage” to “DS is the annual change of total water storage in the closed catchment” in the revised manuscript (Page 5, 172-173.).
Point 21: Line 204-206: Redundancy.
Response 21: Thanks for the suggestion. We changed Line 204-208 to “According to Wang et al. [45], assuming that each single variable is independent in the Budyko-type equations (see Table 2), the elasticity method is used to analyze the contributions of climate and vegetation changes to Eat change. And the first-order approximation of the changes in Eat can be expressed as” in the revised manuscript (Page 6, 215-218.).
References:
Wang, W.; Zou, S.; Shao, Q.; Xing, W.; Chen, X.; Jiao, X.; Luo, Y.; Yong, B.; Yu, Z. The analytical derivation of multiple elasticities of runoff to climate change and catchment characteristics alteration. J. Hydrol. 2016, 541, 1042–1056.
Point 22: Line 228: Unnecessary references.
Response 22: Thanks for the suggestion. First, the Mann-Kendall test was proposed by H.B. Mann and M.G. Kendall, which is the basic of the nonparametric trend-free prewhitening Mann-Kendall test. Then, in order to eliminate the effect of a serial correlation on the trend value. Yue et al improved the Mann-Kendall method and proposed an alternative approach, namely, the nonparametric trend-free prewhitening Mann-Kendall test in 2002. Since the nonparametric trend-free prewhitening Mann-Kendall test was used in this study, we presented all the references in the manuscript.
References:
Mann, H.B. Non-parametric tests against trend. Econometrica. 1945, 13, 245-259.
Kendall, M.G. Rank Correlation Measure. Charles Griffin. London, UK. 1975.
Yue, S.; Pilon, P.; Cavadias, G. Power of the Mann-Kendall and Spearman’s rho tests for detecting monotonic trends in hydrological series. J. Hydrol. 2002, 259, 254–271.
Yue, S.; Wang, C.Y. Applicability of prewhitening to eliminate the influence of serial correlation on the Mann-Kendall test. Water Resour. Res. 2002, 38.
Point 23: Figure 2a: Not a correct way of showing precipitation.
Response 23: Thanks for the suggestion. We changed the Figure 2a in the revised manuscript. Please see the new figure in the revised manuscript.
Point 24: Figure 2d, 2e: Does not required, it can be merged all in one figure.
Response 24: Thanks for the suggestion. We combined Figures 3b, c and d into one figure in the revised manuscript. Please see the new figure in the revised manuscript.
Point 25: Figure 2f: Does not make sense.
Response 25: Thanks for the concern. In fact, Figure 2f, which indicated the variation trend of sunshine duration (S), is very important for quantifying the contribution rate of St change to actual evapotranspiration change in this study. Specifically, St is an essential element in equation (12), which is the basis for the contribution analysis in Section 4.3. Therefore, we analysed the variation trend of S in this section.
Point 26: Figure 2h: What is the reason of increasing NDVI?
Response 26: Thanks for the concern. In fact, the reason of increasing NDVI is complex, as analysed in Section 2, the changes of six major land use/cover during the study period are not obvious, it indicates that the increase in NDVI is a result of the changes in land use/cover (such as afforestation, returning farmland to forests and grass land. Additionally, Wang et al. (2006) found that the increase in agricultural production and climate change are the main reasons for the NDVI increase in the study area.
References:
Wang, Z., Chen, X., Li, Y. Spatio-temporal changes of NDVI in the Pearl River Basin. Ecologic Science Aug. 2006, 25(4), 303-307. (In Chinese)
Point 27: Figure 5: All three figures seems close to each other and doesn’t represent the variation of RMSE.
Response 27: Sorry about the mistake. The RMSE in Figure 5a should be 7.43, we corrected the value of RMSE in the Figure 5a in the revised manuscript.
Point 28: Line 386-387: How much? No value of stat, provide to see the contribution!
Response 28: Thanks for the suggestion. We added the statistic data of contribution rate for climate change and vegetation change to actual evapotranspiration in the conclusion (Page 14, 415-416.). The results shows that climate change contributed (whose average contribution is 149.6%) more to actual evapotranspiration change than vegetation change (which contribution rate is -49.4%).

Reviewer 3 Report
Overall estimate: The paper is interesting, well organized and deserves to be published after minor review. The authors should consider the following points:
In the abstract (line 20) the parameter n should be described a little bit In figure 1, elevation is without units, this information should be given More discussion must be included regarding the limitations and usefulness of the proposed approach. Also, the transferability to other case studies with different climatological conditions should be noted.Author Response
Response to Reviewer 3 Comments
Point 1: In the abstract, the parameter n should be described a little bit.
Response 1: Thanks for the concern. The single parameter n in the Budyko-type equations is usually used to reflect the land surface characteristics. We have revised it in the abstract (Page 1, Line 21-22.).
Point 2: In figure 1, elevation is without units, this information should be given.
Response 2: Sorry for the mistake. The elevation in figure 1 is m. We have provided the revised figure in the revised manuscript.
Point 3: More discussion must be included regarding the limitations and usefulness of the proposed approach. Also, the transferability to other case studies with different climatological conditions should be noted.
Response 3: Thanks for the concern. This study aimed to quantify the impacts of climate change and vegetation variations on regional actual evapotranspiration (Ea). First, by applying an 11-year time window method for each variable, an empirical formula for the parameter n in the Budyko-type equations was established with three basic climatic variables and the NDVI by using stepwise linear regression. Then, elasticity and contribution analyses were performed to quantify the contribution of different factors to Ea variability. The results of this study will provide a basis for understanding the impacts of changes in climate and vegetation on water and energy cycles, improving the accuracy of hydrological forecasting and developing a suitable strategy for water resource management. However, the proposed empirical formulas for the parameter nt in the Budyko-type equations (show in following) still has some limitations.
As shown in equation (11a-11c), the coefficients of each variables in these equations were determined by the nt values, which estimated based on the observed data sets of runoff and other meteorological elements in the study area using each Budyko-type equation. Therefore, the coefficient of each variable in equation (11a-11c) is depend on the characteristics of the climate and land surface conditions in the North and South Panjiang basin. In other regions or catchments that with different climatological conditions, the coefficient of each variable in equation (11a-11c) will inevitably change, this should be further studied in the future. However, the empirical formula for the parameter nt proposed in this study is a working example for other catchments with different climate and vegetation conditions. In further studies, we can throng establishing the relationship between parameter n and the factors of climate and vegetation to investigate the impacts of climate change and vegetation variation on the water cycle of the catchment. We have added the limitations in the revised manuscript (Page 9, Line 284-286.).

Reviewer 4 Report
This study by Li et al. aims at quantifying the effects of climate change and vegetation change on evaporation in the North-South Panjiang basin. The authors use one-parameter Budyko models, and treat the basin-specific parameter as a dynamic variable. The value of this variable is estimated from climatic variables, averaged over a moving window of 11 years. An elasticity and contribution analysis is then performed on the Budyko model with time-varying parameter.
The Budyko framework has been shown in many studies to be an appropriate framework for contribution analysis. Furthermore, the effect of different factors on the basin-specific parameter is an important question.
My main concern with this manuscript is about novelty. Indeed, the authors motivate the study by stating (l. 83 and following) that the basin-specific parameter is usually treated as a constant, and that a “new method” (l. 86) is necessary. However, there are some studies which treat the Budyko parameter dynamically, e.g. Tian et al (2018) and Liu et al. (2016), as well as Jiang et al. (ref. 11 in the manuscript). Please include these in the introduction and show the novelty of the presented approach compared to the published ones. Regarding elasticity of evaporation and contribution of different (climatic and land-related) factors to changes in water balance, I would ask the authors to present in more detail in the Introduction which methods have been used (including the framework used in the manuscript) and what has been found. Against this background, please state what is new in this manuscript.
Further comments:
Minor comments:
Throughout the text, the authors use the term “variability”. I am not sure that this is always the appropriate term, as variability usually describes the spread within a dataset, which is often not what is meant here. Maybe “change” is more appropriate in some situations?
l.46: “have increase in the interest” => “have increased interest”
L.67: necessary for what?
L.72: “Conceptual appeal” is quite vague, please explain.
L.76-77: “basin specific” => “basin-specific”
L.96: “To improving” => “To improve”
L.115-117: As you do not present any data on the water cycle yet, you cannot make the claim that the impacts of LU/LC were “very weak”. Please provide evidence or rephrase this statement.
L.122: Remove “which”.
l.124: The word “mainly” makes it seem that the enumeration is incomplete. Please clarify
l.139 and following: Do the land use/cover changes refer to the values given in Table 1? If so, please provide a cross-reference.
L.142 and Table 1: In my understanding, “construction land” is a planning term and refers to land for which construction is foreseen. Maybe “built-up” is more appropriate?
l.168: “In practical applicationS” and “of *the* moving window method”
l.200 : Please specify how the value of n is calculated using the Budyko-type equations (I suppose through fitting/calibrating, but please state this explicitly).
L.212: It seems that the methodology for Eq. 7 is presented in Ref. 44. If so, please indicate this reference here as well.
L.233: “S presents *a* slight decreasing trend”
L.255: What do you mean by “significantly”? If this is not about statistical significance, I would suggest choosing another term. Also, what is the rationale for not keeping the full model although it gives equal or better R2?
L.257: “modeled” => “model”
Around l. 269: please explain briefly what a lower or higher value of n means for water balance
L.275: “*the* sign”
Fig. 3: the values on the x-axis are the values against which the regression has been performed, right? If so, please specify this, as this allows for a better interpretation of these results.
L.275: Is the Hai basin hydro-climatically similar to the basin studied here? Without this knowledge, it is difficult to know what this result means in the context of this study.
L.282: “formal” => “formula”
L.283: “validate”: see my comment above regarding Fig. 4.
l.299: “Wang-Tang”
L.313: “Attribution” has a quite specific meaning in the context of climate change, which I am not sure is meant here. Please define what is meant here.
L.322: “Is more sensitivity” => “Is more sensitive”
References:
Liu, Q. et al. (2016): The hydrological effects of varying vegetation characteristics in a temperate water-limited basin: Development of the dynamic Budyko-Choudhury-Porporato (dBCP) model. Journal of Hydrology https://doi.org/10.1016/j.jhydrol.2016.10.035
Tian, L. et al (2018): Quantifying the Impact of Climate Change and Human Activities on Streamflow in a Semi-Arid Watershed with the Budyko Equation Incorporating Dynamic Vegetation Information. Water https://doi.org/10.3390/w10121781
Author Response
Response to Reviewer 4 Comments
Main comments:
Point 1: My main concern with this manuscript is about novelty. Indeed, the authors motivate the study by stating (l. 83 and following) that the basin-specific parameter is usually treated as a constant, and that a “new method” (l. 86) is necessary. However, there are some studies which treat the Budyko parameter dynamically, e.g. Tian et al (2018) and Liu et al. (2016), as well as Jiang et al. (ref. 11 in the manuscript). Please include these in the introduction and show the novelty of the presented approach compared to the published ones. Regarding elasticity of evaporation and contribution of different (climatic and land-related) factors to changes in water balance, I would ask the authors to present in more detail in the Introduction which methods have been used (including the framework used in the manuscript) and what has been found. Against this background, please state what is new in this manuscript.
Response 1: Thanks for the concern. In recent years, some studies found that the time-varying parameter in the Budyko framework can better describe the changing processes of Ea and separate the impacts of climate change and other land-related factors on Ea change in the changing environment, such as Jiang et al. (2015), Liu et al. (2016) and Tian et al. (2018). At the same time, combing the elasticity method and Budyko framework (such as equation proposed by Turc-Pike et al. (1964), Fu (1981), Zhang et al., (2001), Yang et al., (2008) and Wang-Tang (2014)), some studies specifically explored the impacts and contributions of climatic factors and land-related indices on the evapotranspiration change (e.g., Yang and Yang (2011), Jiang et al. (2015), Xu et al. (2014) and Wang et al. (2016)). However, with the time-varying parameter in the Budyko-type equations, limited studies have specifically investigated the impacts and contributions of different numbers of climatic factors and vegetation indices on the Ea change by using the elasticity method. Therefore, we try to establish the relationship between the time-varying parameter n and temporally changing variables and further quantify each related climatic factors and vegetation indices on the Ea by using the elasticity method in this study. Thus, this study attempt to develop a method to specifically investigate the contributions of different numbers of climatic factors (which includes precipitation, maximum temperature, minimum temperature, sunshine duration, wind speed and relative humidity) and vegetation indices (NDVI) by combining the time-varying parameter n in Budyko-type equations and elasticity method, because previous studies with time-varying parameter n in Budyko-type equations can not specifically explored the impacts of different climatic factors on Ea change. Based on this, we revised the Introduction in the revised manuscript (Page 2, Line 84-95.).
References:
Jiang, C.; Xiong, L.; Wang, D.; Liu, P.; Guo, S.; Xu, C.Y. Separating the impacts of climate change and human activities on runoff using the Budyko-type equations with time-varying parameters. J. Hydrol. 2015, 522, 326–338.
Liu, Q.; McVicar, T.R.; Yang, Z.; Donohue, R.J.; Liang, L.; Yang, Y. The hydrological effects of varying vegetation characteristics in a temperate water-limited basin: Development of the dynamic Budyko-Choudhury-Porporato (dBCP) model. J. Hydrol. 2016, 543, 595–611
Tian, L.; Jin, J.; Wu, P.; Niu, G. Quantifying the impact of climate change and human activities on streamflow in a Semi-Arid watershed with the Budyko equation incorporating dynamic vegetation information. Water (Switzerland) 2018, 10.
Pike, J.C. The estimation of annual runoff from meteorological data in a tropical climate. J. Hydrol. 1964, 2, 116-123.
Fu, B.P. The calculation of the evaporation from land surface. Scientia Atmospherica Sinica. 1981, 5, 23-31. (In Chinese)
Zhang L.; Dawes, W.R.; Walker, G.R. Response of mean annual evapotranspiration to vegetation changes at catchment scale. Water Resour. Res. 2001, 37, 701.
Yang, H.; Yang, D.; Lei, Z.; Sun, F. New analytical derivation of the mean annual water-energy balance equation. Water Resour. Res. 2008, 44, 1–9.
Wang, D.; Tang, Y. A one-parameter Budyko model for water balance captures emergent behavior in darwinian hydrologic models. Geophys. Res. Lett. 2014, 41, 4569–4577.
Yang, H.; Yang, D. Derivation of climate elasticity of runoff to assess the effects of climate change on annual runoff. Water Resour. Res. 2011, 47, 1–12.
Xu, X.; Yang, D.; Yang, H.; Lei, H. Attribution analysis based on the Budyko hypothesis for detecting the dominant cause of runoff decline in Haihe basin. J. Hydrol. 2014, 510, 530–540.
Wang, W.; Zou, S.; Shao, Q.; Xing, W.; Chen, X.; Jiao, X.; Luo, Y.; Yong, B.; Yu, Z. The analytical derivation of multiple elasticities of runoff to climate change and catchment characteristics alteration. J. Hydrol. 2016, 541, 1042–1056.
Point 2: Please provide some additional context on the three Budyko equations selected for this study. What is the rationale behind them? Do they differ in their assumptions?
Response 2: Thanks for the concern. First of all, the three Budyko-type equations (proposed by Fu, Zhang et al. and Wang-Tang) selected in this study are all based on the assumption proposed by Budyko (1974), which indicated that the evapotranspiration is limited by available water (i.e., precipitation) and available energy (i.e., potential evaporation). Specifically, the equation proposed by Fu states that for a given potential evaporation, the rate of the change in catchment evaporation with respect to precipitation ( ) increases with residual potential evaporation (EO - P) but decreases with precipitation (P). Similarly, for a given precipitation the rate of the change in evaporation with respect to potential evaporation ( ) increases with residual precipitation (P - Ea) but decreases with potential evaporation (EO) (Zhang et al., 2008). The equation proposed by Zhang et al. states there is a certain functional relationship between Ea/P and Eo / P based on the influence of climatic conditions (Zhang et al., 2001). Wang-Tang derived a one-parameter Budyko-type equation based on a generalization of the proportionality hypothesis of the SCS model and it is equivalent to the key equation of the “abcd” model. The obvious difference of Wang-Tang’s equation is that it illustrated a temporal pattern of water balance amongst Darwinian hydrologic models (Budyko framework, SCS), which allows for synthesis with the Newtonian approach (the individual physical processes acting in a watershed) and offers opportunities for progress in hydrologic modelling (Wang-Tang, 2014). Additionally, we chose these three Budyko-type equations mainly because their universal applicability and the fact that it requires only routinely recorded weather data.
Reference:
Fu, B.P. The calculation of the evaporation from land surface. Scientia Atmospherica Sinica. 1981, 5, 23-31. (In Chinese)
Zhang L.; Dawes, W.R.; Walker, G.R. Response of mean annual evapotranspiration to vegetation changes at catchment scale. Water Resour. Res. 2001, 37, 701.
Wang, D.; Tang, Y. A one-parameter Budyko model for water balance captures emergent behavior in darwinian hydrologic models. Geophys. Res. Lett. 2014, 41, 4569–4577.
Budyko M.I. Climate and life. Academic, San Diego, CA. 1974.
Zhang, L.; Potter, N.J.; Zhang, Y. Water balance modeling over variable time scales based on the Budyko framework - Model development and testing. J. Hydrol. 2008, 390, 121–122.
Point 3: The window size selected is 11 years. Tian et al. (2018) compared different window size and found that the estimated Budyko parameter did not change much for window sizes larger than 13 years. It would be helpful to discuss this point, either in the Methods section (for the choice of window size) or in the Discussion. Also, it is not quite clear how the choice of window size relates to the length of the time series (l. 172).
Response 3: Thanks for the suggestion. Firstly, the study proposed by Zhang et al. (2004, 2008) suggested that if the study period is long enough, i.e. decades, the changes of water storage term in the water balance equation can be neglected. And then, Jiang et al. (2015) and Tian et al. (2018) used 11 years and 13 years window to remove the influences of the water storage in the water balance equation. In this study, we found that the results of precipitation minus runoff did not change much for time window size larger than 11 years, while 11 years window is closed to the window size used by Jiang et al. (2015) and Tian et al. (2018). Thus, we decided to select 11-year time window in this study. Based on this, we changed “Based on the length of the hydrological and meteorological series in the NSPB, an 11-year time window is applied to annual R, Ea, P and DS series” to “According to previous studies [13, 38, 40], the changes of total water storage can be neglected during a long time period, different sizes of time windows, such as 11 year, 13 years, were used in studies proposed by Jiang et al. [13] and Tian et al. [40], respectively. Here, as the P-R did not change much for time window size larger than 11 years, we selected the 11-year time window to minimize the influence of changes in DS series” (Page 5, Line 178-183.).
References:
Zhang L.; Hickel, K.; Dawes, W.R.; Western, A.W.; Chiew, F.H.S.; Briggs, P.R. A rational function approach for estimating mean annual evapotranspiration. Water Resour. Res. 2004, 40, 1–14.
Zhang, L.; Potter, N.J.; Zhang, Y. Water balance modeling over variable time scales based on the Budyko framework - Model development and testing. J. Hydrol. 2008, 390, 121–122.
Jiang, C.; Xiong, L.; Wang, D.; Liu, P.; Guo, S.; Xu, C.Y. Separating the impacts of climate change and human activities on runoff using the Budyko-type equations with time-varying parameters. J. Hydrol. 2015, 522, 326–338.
Tian, L.; Jin, J.; Wu, P.; Niu, G. Quantifying the impact of climate change and human activities on streamflow in a Semi-Arid watershed with the Budyko equation incorporating dynamic vegetation information. Water (Switzerland) 2018, 10.
Point 4: A major caveat of the results presented in Fig. 4 is that the models were applied to the same dataset from which the parameter values were estimated. This is probably dictated by data availability. While the comparison of time series has some value and shows to what extent the use of a dynamic n improves the Ea estimations, I would be very hesitant to call this a “validation”.
Response 4: Thanks for the suggestion. We changed “the empirical formula for nt is used together with the Budyko-type equations to validate the values of Eat in the NSBP” to “the empirical formula for nt is used together with the Budyko-type equations to calibrate the values of Eat in the NSBP” in the revised manuscript (Page 9, Line 292).
Minor Comments:
Point 1: Throughout the text, the authors use the term “variability”. I am not sure that this is always the appropriate term, as variability usually describes the spread within a dataset, which is often not what is meant here. Maybe “change” is more appropriate in some situations?
Response 1: Thanks for the suggestion. We replaced "variability" with "change" where needed. The specific changes are highlighted in yellow in the revised manuscript.
Point 2: L.46: “have increase in the interest” => “have increased interest”
Response 2: Thanks for the suggestion. We corrected "have increase in the interest" to "have increased interest" in the revised manuscript (Page 1, Line 45).
Point 3: L.67: necessary for what?
Response 3: Thanks for the concern. We changed the expression to “Previous studies have suggested that it is necessary to explicitly incorporate key indices of vegetation conditions into the Budyko framework for many catchments.” in the revised manuscript (Page 2, Line 66-68).
Point 4: L.72: “Conceptual appeal” is quite vague, please explain.
Response 4: Thanks for the concern. As mentioned in Line 153-156, in general, the water-energy balance for a catchment over a long-term timescale describe the relationship between the components of water and heat balances of land [Budyko, 1974, p. 322]. Based on this, Budyko postulated that the primary factors controlling the rate of long-term average evapotranspiration (Ea) are the availability of energy and water. Usually, the potential evaporation (Eo) is used to measure the availability of energy, while precipitation (P) is used to measure the availability of water (Yang et al., 2009). The Budyko hypothesis can be expressed as:
where the function F was supposed to have a common form. In addition, this framework has been proved by many scholars, such as Fu et al. (1981), Zhang et al. (2001, 2004 and 2008), Yang et al. (2008), Wang-Tang (2014).
References:
Yang, D.; Shao, W.; Yeh, P.J.F.; Yang, H.; Kanae, S.; Oki, T. Impact of vegetation coverage on regional water balance in the nonhumid regions of China. Water Resour. Res. 2009, 45, 1–13.
Fu, B.P. The calculation of the evaporation from land surface. Scientia Atmospherica Sinica. 1981, 5, 23-31. (In Chinese)
Zhang L.; Dawes, W.R.; Walker, G.R. Response of mean annual evapotranspiration to vegetation changes at catchment scale. Water Resour. Res. 2001, 37, 701.
Zhang L.; Hickel, K.; Dawes, W.R.; Western, A.W.; Chiew, F.H.S.; Briggs, P.R. A rational function approach for estimating mean annual evapotranspiration. Water Resour. Res. 2004, 40, 1–14.
Zhang, L.; Potter, N.J.; Zhang, Y. Water balance modeling over variable time scales based on the Budyko framework - Model development and testing. J. Hydrol. 2008, 390, 121–122.
Wang, D.; Tang, Y. A one-parameter Budyko model for water balance captures emergent behavior in darwinian hydrologic models. Geophys. Res. Lett. 2014, 41, 4569–4577.
Point 5: L.76-77: “basin specific” => “basin-specific”
Response 5: Thanks for the suggestion. We corrected "basin specific" to "basin-specific" in the revised manuscript (Page 2, Line 77-78).
Point 6: L.96: “To improving” => “To improve”
Response 6: Sorry for the mistake. We corrected "To improving" to "To improve" in the revised manuscript (Page 2, Line 104).
Point 7: L.115-117: As you do not present any data on the water cycle yet, you cannot make the claim that the impacts of LU/LC were “very weak”. Please provide evidence or rephrase this statement.
Response 7: Thanks for the suggestion. We corrected " which indicates that the impacts of land use/cover change on the water cycle were very weak from 1980-2010 in this catchment" to "thus, it can be assumed that the land use/cover in the NSPB are almost constant from 1980 to 2010" in the revised manuscript (Page 3, Line 123-124.).
Point 8: L.122: Remove “which”.
Response 8: Thanks for the suggestion. We revised to " Tian-e hydrologic station is the outlet of the NSPB, which with the average runoff about 450 mm during 1982-2013, and the annual runoff data from 1982 to 2013 at this outlet were obtained from the Ministry of Water Resources of China in this study ." in the revised manuscript (Page 4, Line 129-131).
Point 9: L.124: The word “mainly” makes it seem that the enumeration is incomplete. Please clarify
Response 9: Thanks for the suggestion. We removed "mainly" in the revised manuscript (Page 4, Line 132).
Point 10: L.139 and following: Do the land use/cover changes refer to the values given in Table 1? If so, please provide a cross-reference.
Response 10: Yes, the land use/cover changes refer to the values given in Table 1. We added the reference (Yang et al. 2009) in the revised manuscript (Page 4, Line 150).
References;
Yang, D.; Shao, W.; Yeh, P.J.F.; Yang, H.; Kanae, S.; Oki, T. Impact of vegetation coverage on regional water balance in the nonhumid regions of China. Water Resour. Res. 2009, 45, 1–13.
Point 11: L.142 and Table 1: In my understanding, “construction land” is a planning term and refers to land for which construction is foreseen. Maybe “built-up” is more appropriate?
Response 11: Thanks for the suggestion. We corrected "construction land" to "built-up land" in the revised manuscript (Page 4, Line 150 and Table 1.).
Point 12: L.168: “In practical application” and “of *the* moving window method”
Response 12: Thanks for the suggestion. We corrected "In practical application of moving window method" to "Specially, in the application of moving window method" in the revised manuscript (Page 5, Line 176.).
Point 13: L. 200: Please specify how the value of n is calculated using the Budyko-type equations (I suppose through fitting/calibrating, but please state this explicitly).
Response 13: Thanks for the concerns. Firstly, we can calculate the values of Eat by water balance equation (equation (3)) with observed Pt and Rt. And then, we use the obtained values of Eat to calculate the parameter nt based on the Budyko-type equations using the observed Pt and the values of Eot, which calculated by FAO Penman-Monteith equation with observed climatic elements. Based on these two steps, we can obtained the values of nt using the Budyko-type equations.
Point 14: L.212: It seems that the methodology for Eq. 7 is presented in Ref. 44. If so, please indicate this reference here as well.
Response 14: Thanks for the suggestion. We added the reference of Wang et al. (2016) in the revised manuscript (Page 6, Line 216).
References:
Wang, W.; Zou, S.; Shao, Q.; Xing, W.; Chen, X.; Jiao, X.; Luo, Y.; Yong, B.; Yu, Z. The analytical derivation of multiple elasticities of runoff to climate change and catchment characteristics alteration. J. Hydrol. 2016, 541, 1042–1056.
Point 15: L.233: “S presents *a* slight decreasing trend”
Response 15: Thanks for the suggestion. We corrected "S presents slight decreasing trend" to "S presents a slight decreasing trend" in the revised manuscript (Page 7, Line 243.).
Point 16: L.255: What do you mean by “significantly”? If this is not about statistical significance, I would suggest choosing another term. Also, what is the rationale for not keeping the full model although it gives equal or better R2?
Response 16: Thanks for the suggestion. We corrected "significantly" to "effectively" in the revised manuscript (Page 8, Line 264.). In general, a good final fitted model of the parameter n should be simple in form while achieving satisfactory simulation accuracy. Therefore, when the value of r2 is close or equal, we chose a simple model with fewer variables, rather than a complex model that includes all the variables.
Point 17: L.257: “modeled” => “model”
Response 17: Sorry for the mistake. We corrected "fitted modelled" to "fitted model" in the revised manuscript (Page 8, Line 266.).
Point 18: Around l. 269: please explain briefly what a lower or higher value of n means for water balance
Response 18: Thanks for the concern. In Budyko framework, the parameter n mainly affects the water balance by affecting the relationship between Ea / P and Eo / P. Usually, the higher value of n means the higher value of Ea / P while the value of Eo / P remains the same. Specifically, when the value of P is constant, the higher value of n means the higher value of Ea while the value of Eo remains the same (see Figure 6 in the studies proposed by Yang et al. 2009.).
References;
Yang, D.; Shao, W.; Yeh, P.J.F.; Yang, H.; Kanae, S.; Oki, T. Impact of vegetation coverage on regional water balance in the nonhumid regions of China. Water Resour. Res. 2009, 45, 1–13.
Point 19: L.275: “*the* sign”
Response 19: Thanks for the suggestion. We corrected "sign" to "the sign" in the revised manuscript (Page 9, Line 282.).
Point 20: Fig. 3: the values on the x-axis are the values against which the regression has been performed, right? If so, please specify this, as this allows for a better interpretation of these results.
Response 20: Thanks for the concern. In Figure 3, the x-axis presents the values of , and , which derived from the Budyko-type equations using the observations, the y-axis presents the values of , and , which estimated by equation (11a-11c) proposed in this study.
Point 21: L.275: Is the Hai basin hydro-climatically similar to the basin studied here? Without this knowledge, it is difficult to know what this result means in the context of this study.
Response 21: Thanks for the suggestion. In fact, the hydro-climatic conditions exist some differences between Hai River Basin and the NSPB, because the Hai River basin located in northern China, while the NSPB located in southern China. The reason we presented the results proposed by Yang et al. (2009) is that no studies have been performed to investigate the changes of Ea based on the Budyko-type equations in the NSPB, and most studies indicated the positive relationship between Vegetation index (e.g., Feng et al., 2016; Yang et al., 2009; Liang et al., 2015) and the parameter n in China. The negative relationship between NDVI and the parameter n shown in NSPB and the Hai River Basin is a special case, which is interesting to investigate the causes of it. Overall, we finally decided to remove the comparison between the results obtained in this study with the results proposed by Yang et al. (2009) in the revised manuscript (Page 8, Line 282.).
References:
Yang, D.; Shao, W.; Yeh, P.J.F.; Yang, H.; Kanae, S.; Oki, T. Impact of vegetation coverage on regional water balance in the nonhumid regions of China. Water Resour. Res. 2009, 45, 1–13.
Feng, X.; Fu, B.; Piao, S.; Wang, S.; Ciais, P.; Zeng, Z.; Lü, Y.; Zeng, Y.; Li, Y.; Jiang, X.; et al. Revegetation in China’s Loess Plateau is approaching sustainable water resource limits. Nat. Clim. Chang. 2016, 6, 1019–1022.
Liang, W.; Bai, D.; Wang, F.; Fu, B.; Yan, J.; Wang, S.; Yang, Y.; Long, D.; Feng, M. Quantifying the impacts of climate change and ecological restoration on streamflow changes based on a Budyko hydrological model in China’s Loess Plateau. Water Resour. Res. 2015, 51, 6500–6519.
Point 22: L.282: “formal” => “formula”
Response 22: Sorry for the mistake. We corrected "formal" to "formula" in the revised manuscript (Page 9, Line 291.).
Point 23: L.299: “Wang-Tang”
Response 23: Sorry for the mistake. We corrected "Wang" to "Wang-Tang" in the revised manuscript (Page 10, Line 308.).
Point 24: L.313: “Attribution” has a quite specific meaning in the context of climate change, which I am not sure is meant here. Please define what is meant here.
Response 24: Thanks for the suggestion. We changed “analyze the attribution of the Eat change” to “analyze the contribution of climate change and vegetation variations to Eat change” in the revised manuscript (Page 11, Line 322-323.).
Point 25: L.322: “Is more sensitivity” => “Is more sensitive”
Response 25: Sorry to the mistake. We corrected "Is more sensitivity" to "Is more sensitive" in the revised manuscript (Page 11, Line 332.).

Round 2
Reviewer 2 Report
Dear Authors,
Thanks for considering the comments.